# Probing the molecular structure at graphite–water interfaces by correlating 3D-AFM and SHINERS

Lalith Krishna Samanth Bonagiri [1,2,6], Diana M. Arvelo [3,6], Fujia Zhao [1,4,6], Jaehyeon Kim [1,4], Qian Ai [1,4], Shan Zhou [1,4], Kaustubh S. Panse [1,4], Ricardo Garcia [3] ✉ & Yingjie Zhang [1,4,5] ✉

Water at solid surfaces is key for many processes ranging from biological signal transduction to membrane separation and renewable energy conversion. However, under realistic conditions, which often include environmental and surface charge variations, the interfacial water structure remains elusive. Here we overcome this limit by combining three-dimensional atomic force microscopy (3D-AFM) and interface-sensitive shell-isolated nanoparticle enhanced Raman spectroscopy (SHINERS) to characterize the graphite–water interfacial structure in situ. Through correlative analysis of the spatial liquid density maps and vibrational peaks within ≈2 nm of the graphite surface, we find the existence of two interfacial configurations at open circuit potential, a transient state where pristine water exhibits strong hydrogen bond (H-bond) breaking effects, and a steady state with hydrocarbons dominating the interface and weak H-bond breaking in the surrounding water. At sufficiently negative potentials, both states transition into a stable structure featuring pristine water with a broader distribution of H-bond configurations. Our three-state model resolves many long-standing controversies on interfacial water structure.

Water is essential in biology and in technology. Many water-mediated processes depend critically on the structure of solid–water interfaces, such as the electrode–water interface in electrochemical systems, membrane–water interface of biological cells, and the semiconductor–water interface in biosensing and photocatalysis[1–4]. Bulk water has a well-known configuration in the form of a dynamic H-bond network[5]. In contrast, the structure of water near a solid surface (interfacial water) has been a subject of intense debates and controversies. Even the description of ideal interfaces, such as pure water next to a crystalline solid surface, is a challenge for state-of-the-art atomistic simulations[1,3]. In real life, various salt and organic species, either intentionally added or adventitious, inevitably exist in water. If

the concentration of these impurities is low, the bulk water structure is expected to remain unaffected[5]. However, at the interfacial region (within ≈2 nm from a solid surface), even trace amounts of salt or organic species from the bulk may adsorb or accumulate to a high concentration[1,6–8]. As a result, the solid–water interfacial structure is influenced by several factors, including the steric hard wall effect, hydrophobicity, electric field, and surface adhesion. The large variety of possible molecular species and interactions makes the solid–water interface inherently complex.

In the past three decades, many techniques have been developed or implemented to characterize the interfacial water structure[6,9–22]. These methods can be broadly classified into three categories:

[1]Materials Research Laboratory, University of Illinois, Urbana, IL, USA. [2]Department of Mechanical Science and Engineering, University of Illinois, Urbana, IL, USA. [3]Instituto de Ciencia de Materiales de Madrid, CSIC, Madrid, Spain. [4]Department of Materials Science and Engineering, University of Illinois, Urbana, IL, USA. [5]Beckman Institute for Advanced Science and Technology, University of Illinois, Urbana, IL, USA. [6]These authors contributed equally: Lalith Krishna Samanth Bonagiri, Diana M. Arvelo, Fujia Zhao. ✉e-mail: r.garcia@csic.es; yjz@illinois.edu

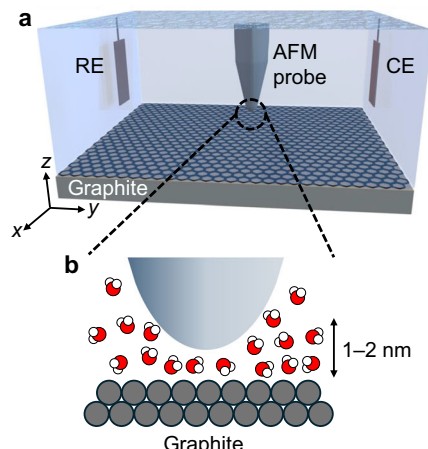

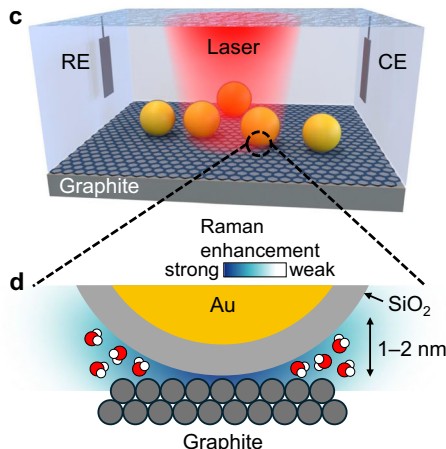

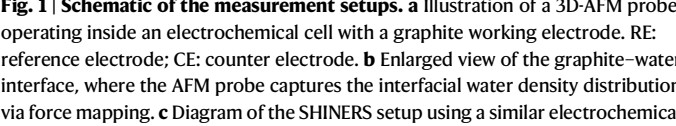

**Fig. 1 | Schematic of the measurement setups. a** Illustration of a 3D-AFM probe operating inside an electrochemical cell with a graphite working electrode. RE: reference electrode; CE: counter electrode. **b** Enlarged view of the graphite–water interface, where the AFM probe captures the interfacial water density distribution via force mapping. **c** Diagram of the SHINERS setup using a similar electrochemical cell. Au/SiO$_2$ core/shell nanoparticles are dispersed on the graphite electrode surface. A laser source is used to induce Raman scattering. **d** Expanded view of the interfacial region, where the Raman signal of the water molecules within 1–2 nm from the graphite surface is enhanced.

interface-sensitive spectroscopies (e.g., sum frequency generation, X-ray absorption, and surface-enhanced Raman)[17–19,23,24], scattering methods (using X-ray or neutron)[20–22], and real-space imaging by 3D-AFM[6,9–16]. The results have been interpreted via two distinct models. One assumes that pristine water molecules occupy the interface, as supported by a lack of non-water peaks in spectroscopies or a ≈3 Å liquid layer spacing in 3D-AFM or X-ray/neutron scattering (intermolecular distance of bulk water is ≈3 Å)[9,11,13,15,19,20]. The other model hypothesizes that adventitious airborne or liquid-borne species (likely hydrocarbons) dominate the 1–2 nm interfacial region. This non-pristine model is based on the observation of hydrocarbon-related vibrational peaks in spectroscopies, a 4–5 Å interlayer distance identified by 3D-AFM, or a reduction of water density at the interface by as high as ≈90%, as measured by X-ray or neutron reflectivity[6,21,25–27]. These discrepancies cannot be attributed to differences in the solid structure, as they occur for many commonly used substrates including Au, graphite, graphene, and self-assembled monolayers (Supplementary Tables 1–4).

In the presence of surface charges or an electric field, the response of interfacial water structure is also highly controversial, with existing reports claiming markedly different degrees of H-bond reconfiguration or change in the interlayer spacing[12,16–19,28] (Supplementary Tables 1–4).

The above controversies in interfacial water structure may stem from several factors. One possibility is the variation in sample conditions across different labs, influenced by differences in water purity, preparation methods, and/or ambient conditions. Another factor is the limited amount and type of information that can be extracted from each individual characterization method; these limits make it difficult to accurately determine interfacial structures using any single technique.

To resolve the discrepancies of solid–water interfaces, we combined the atomic-resolution imaging method 3D-AFM (Fig. 1a, b) with SHINERS[16,19,29] (Fig. 1c, d). Among the existing methods to study solid–water interfaces, 3D-AFM and SHINERS are specifically sensitive to the same 1–2 nm thick interfacial liquid region above flat substrates[19,30–32]. The capability of SHINERS to detect graphite–water interfaces is further verified by the observation of the graphite G band and the strong potential-dependence of the overall signal, as shown later in this article. On the other hand, 3D-AFM captures the local liquid density[30,33] without inducing observable substrate deformations (Supplementary Note 1). Therefore, the combination of these two

methods enables a direct and reliable integration of the complementary spatial density profiles (from 3D-AFM) and chemical bonding configurations (from SHINERS), thus providing comprehensive information on the interfacial liquid structure.

Among the existing interfacial systems, carbon–water interfaces are crucial for a wide range of applications, including energy storage, electrocatalysis, biosensing, water desalination, and more[34–37]. Here, we use highly oriented pyrolytic graphite (HOPG) as the model carbon electrode system, which has a flat, homogeneous basal plane surface enabling a reliable comparison of imaging and spectroscopy data. Additionally, the simple mechanical exfoliation method for surface preparation ensures reproducible measurements in different labs. The experiments were carried out independently over a four-year period under different environmental conditions in two locations, Madrid (Spain) and Urbana (IL, USA). A combination of 16 separate sets of 3D-AFM and 4 sets of SHINERS measurements is reported here; each measurement lasted for 1–2 days, during which the potential-dependence and time evolution of interfacial liquids were thoroughly examined across multiple sample spots. A comprehensive analysis of all the data together reveals three types of interfacial configurations: a transient pristine water state with strong H-bond breaking near open circuit potential (OCP) (state 1), a stable hydrocarbon-dominated structure with weak H-bond breaking of water near OCP (state 2), and a stable pristine water state with a broad distribution of H-bond configurations at sufficiently negative potentials (state 3). These states transition between each other due to either time evolution or potential modulation.

## Results and discussion

### Interfacial configurations at OCP

In this work, OCP values for all the liquid electrolytes were similar and determined to be $0.17 \pm 0.15$ V vs. Ag/AgCl. For both 3D-AFM and SHINERS, results measured within the range of 0–0.4 V vs. Ag/AgCl did not show observable differences. Therefore, potentials within 0–0.4 V vs. Ag/AgCl can be regarded as at or near OCP.

All 3D-AFM measurements were performed on atomically flat terrace sites of HOPG immersed in water or aqueous solution (Supplementary Fig. 1). The HOPG surface remained stable over time and after potential cycles (Supplementary Fig. 2). While step edges exist on the HOPG surface[38,39], those sites were not examined by 3D-AFM in this work. For 3D-AFM, we imaged a volume of at least 5 nm × 5 nm × 1.5 nm along $x$, $y$ (in-plane), and $z$ (out-of-plane) directions at each location, to

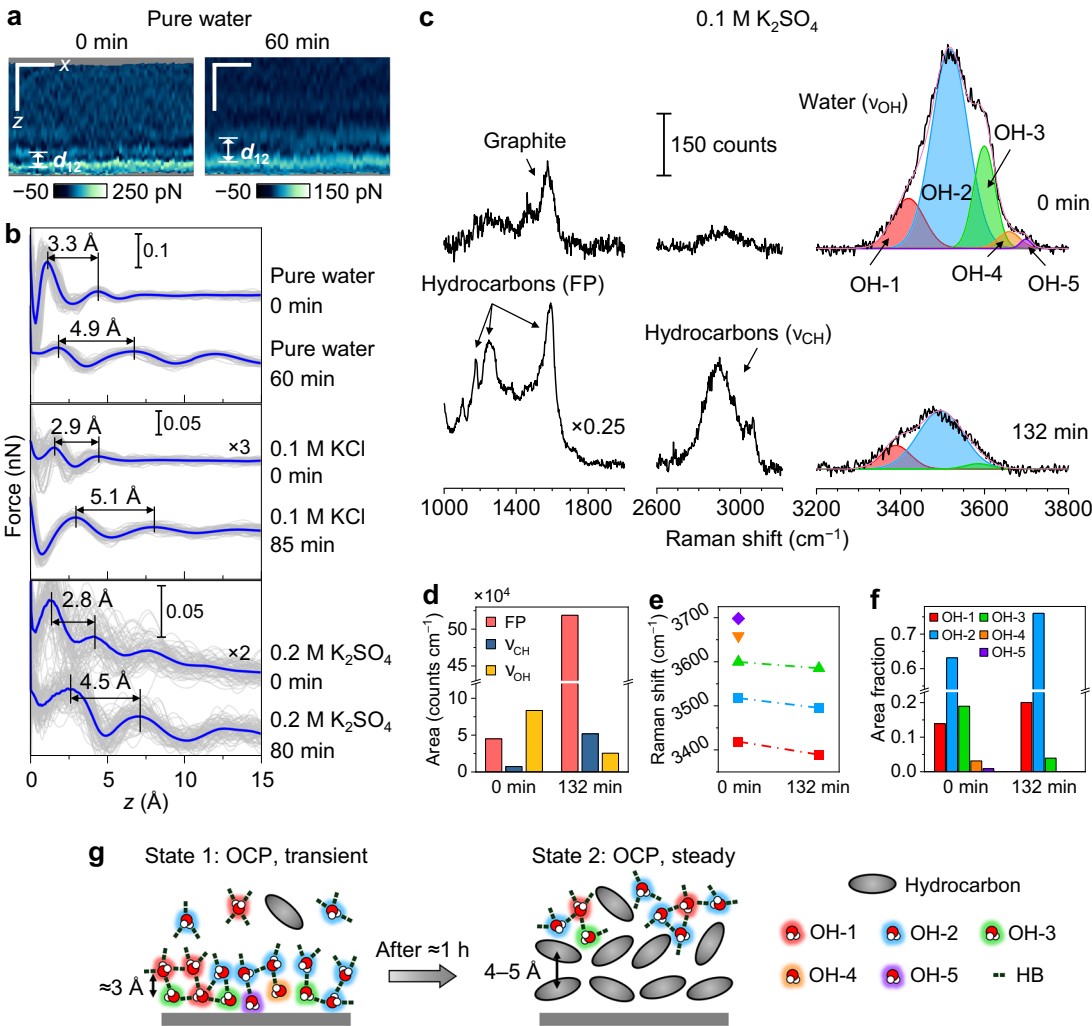

**Fig. 2 | Time-dependent evolution of HOPG/aqueous solution interfaces at OCP.**
**a** 3D-AFM $x$–$z$ force maps of an HOPG/pure water interface obtained at the beginning of the measurement and after 60 min (conducted in Madrid). Scale bars: 1 nm ($x$ and $z$ directions). **b** Force–distance curves of HOPG/pure water (extracted from (**a**)) and HOPG/aqueous solutions interfaces, at the beginning of the experiment and after 1–1.5 h (Madrid). Individual curves are shown in gray, and average curves in blue. The interlayer distance $d_{12}$ is marked on each set of curves. **c** SHINERS spectra of an HOPG/0.1 M $K_2SO_4$ solution interface at the beginning ($t = 0$ min) and after 132 min across three spectral regions: fingerprint (FP), C–H stretching ($\nu_{CH}$), and O–H stretching ($\nu_{OH}$). The $\nu_{OH}$ band is deconvoluted into five

individual Voigt peaks, OH-1 (red), OH-2 (blue), OH-3 (green), OH-4 (orange), and OH-5 (violet). The sum of the fits is shown in pink. **d** Time evolution of the peak areas for FP, $\nu_{CH}$, and $\nu_{OH}$ modes. **e**, **f** Peak positions and area fractions of the individual $\nu_{OH}$ peaks for $t = 0$ and 132 min. **g** Schematic diagrams of the time-dependent evolution of HOPG–water interfaces at OCP, depicting the replacement of interfacial water by adventitious hydrocarbons and associated changes in the hydrogen-bonding environment—specifically, an increase in the fractions of OH-1 and OH-2, a decrease in the fraction of OH-3, and the disappearance of OH-4 and OH-5 configurations. Source data are provided as a Source Data file.

ensure statistical significance. At OCP, $x$–$z$ maps of pure water at the HOPG surface reveal multiple discrete layers, where the interlayer spacing evolved from the initial value of ≈3 Å to 4–5 Å within about 1 h (Fig. 2a, b, Supplementary Fig. 3a–c). The 0 min time point corresponds to the time when the first 3D-AFM data frame was acquired at the pristine HOPG–water interface. The observed time-evolution is consistent with previous reports[6,25,40]. As explained before, ≈3 Å corresponds to pristine hydration layers, while 4–5 Å is likely due to the accumulation of adventitious species whose exact nature remains unknown to date[6,25,40]. The average force vs. $z$ curves extracted from the $x$–$z$ maps (Fig. 2a), shown in Fig. 2b, pure water panel, confirmed the same layered structures with damped oscillations as typical for interfacial liquids[30]. Most of the force curves obtained in this work exhibited at least two peaks under all sample and probe conditions. Whenever more than two peaks were identified, the upper layers showed similar interlayer distances as the 1st to 2nd layer (Supplementary Fig. 4). Therefore, the 1st–2nd layer distance ($d_{12}$) is a reliable

experimental descriptor of the observed interfacial liquid density profile. This distance will be used for statistically significant analysis throughout this work.

For the rest of the article, although full 3D imaging was performed for each AFM result, we only show the individual and average force curves in the figures to avoid redundancy and highlight the evolution of $d_{12}$.

3D-AFM of electrolyte solutions, including $K_2SO_4$ in water (0.01–0.2 M) and KCl in water (0.1 M), revealed similar layered interfacial liquid structures with nearly the same time evolution of $d_{12}$ (Fig. 2b, Supplementary Fig. 3). These results underline the reproducibility of the observed interfacial structural evolution. Furthermore, the findings are largely independent of the salt species and concentration. We thus rule out the possibility that changes in interlayer spacing are induced by interfacial ion aggregation. This interpretation is consistent with prior reports suggesting that hydration layer spacings are largely insensitive to the identity and concentration of the

supporting electrolyte in dilute solutions (<1 M), since the number density of water is much higher than that of the ionic species in these solutions[9,11,30,41,42]. Note that the distance between the first peak and the substrate exhibits large variations among most of the results, which have been previously observed in aqueous solutions[10,14,43] and attributed to the fluctuations in the hydration condition of the end of the AFM probe, rather than the intrinsic liquid structure[44]. $d_{12}$, on the other hand, is robustly reproducible and independent of the AFM probe, as reported in refs. [6,25,30,43], and evident from the data shown in Supplementary Figs. 3–14, Supplementary Tables 5, 6.

Additionally, care must be taken to avoid overinterpreting differences in the observed 3D-AFM force curves. Random fluctuations, due to variations in AFM tip condition, environmental factors and instrument noise (considering the two different locations and four-year time span), can result in various error ranges in the experimental observables. For example, average $d_{12}$ values within 2.8–3.3 Å were observed for pristine HOPG–aqueous solution interfaces with different ionic species and concentrations (Supplementary Fig. 15). Force oscillation amplitudes, on the other hand, exhibits larger fluctuations between 0.01–0.1 nN (Supplementary Fig. 16). These variations should not be directly attributed to differences in the electrolytes or aging/cleanliness conditions.

We further interrogated the chemical composition of the HOPG–water interface using SHINERS. As shown in Fig. 1c, d, the substrate for SHINERS consists of Au/SiO$_2$ core/shell nanoparticles deposited on HOPG (Supplementary Figs. 17, 18). Detailed SHINERS experimental procedures and validation of its non-perturbative nature in probing interfacial liquids were reported in our previous publications[16,45]. The particle–HOPG gap regions (1–2 nm thick) are hotspots for plasmonic enhancement[32]. When immersed in aqueous solutions, these hotspots are filled by the interfacial liquid species, which contribute to the SHINERS signal[19,31]. While bulk solution (more than ≈2 nm away from the electrode surface) also gives rise to a small fraction of the Raman signal, its contribution was subtracted to obtain the presented SHINERS spectra (Supplementary Figs. 19, 20).

The confocal Raman spectrum of bulk solution (0.1 M K$_2$SO$_4$ in water) confirmed the standard O–H stretching ($\nu_{OH}$) band of water molecules between 3000–3700 cm$^{-1}$, as well as the absence of adventitious species within the sensitivity limit (Supplementary Fig. 19). In contrast, SHINERS spectra of an electrochemically cleaned HOPG–solution interface at OCP showed graphite G band[46,47] at 1580 cm$^{-1}$, a weak C–H stretching ($\nu_{CH}$) peak[48,49] centered around 2900 cm$^{-1}$, and $\nu_{OH}$ band with a distinct shape compared to the bulk solution (Fig. 2c, 0 min panels). The $\nu_{CH}$ mode can be attributed to the presence of small amounts of hydrocarbon species at the interface that may be either aliphatic or aromatic in nature[50–52]. This as-prepared HOPG–solution interface can be viewed as pristine, since the amount of hydrocarbons inferred from the spectra is minute. The absence of hydrocarbons confirms the effectiveness of the electrochemical cleaning methods (details explained in the following sections) in removing the possible hydrocarbons at the original HOPG–solution interface and ligands on the Au/SiO$_2$ nanoparticle surface. After ageing for about 2 h, the $\nu_{OH}$ band intensity became lower, the $\nu_{CH}$ peaks became stronger, and a series of peaks emerged in the 1050–1700 cm$^{-1}$ range, which we assign as different modes of hydrocarbon fingerprints (FP) (Fig. 2c, 132 min panels). Possible origins of the FP modes include C–O stretches, C–F stretches, and aromatic ring vibrations, among others, although precise assignments are still under debate[53]. The simultaneous decrease of water peaks ($\nu_{OH}$ peak area) and increase in hydrocarbon peaks (FP and $\nu_{CH}$ peak areas) (Fig. 2d) confirmed the interfacial accumulation of hydrocarbons that replaced the original hydration layers. This conclusion is consistent with the increase of interlayer distance from ≈3 Å to 4–5 Å over time observed in the 3D-AFM results (Fig. 2a, b).

Remarkably, ageing leads to significant changes in the $\nu_{OH}$ peak components at the HOPG–solution interface. The $\nu_{OH}$ band at the pristine interface was decomposed into five peaks, OH-1– OH-5, with increasing peak positions at 3419, 3517, 3600, 3658, and 3698 cm$^{-1}$, respectively (Fig. 2c, 0 min panel, $\nu_{OH}$ region, and Fig. 2e). After ≈2 h of ageing, the first three peaks were red-shifted to 3389, 3495, and 3585 cm$^{-1}$, respectively, while the OH-4 and OH-5 peaks became absent or negligible (Fig. 2c, 132 min panel, $\nu_{OH}$ region, and Fig. 2e). For both pristine and aged interfaces, OH-2 remained the dominant peak, with area fractions of 63% and 76%, respectively; upon ageing, the OH-3 contribution diminished nearly five-fold (from 19% to 4%), while OH-1 became more prominent, increasing from 14% to 20% (Fig. 2f).

The assignment of $\nu_{OH}$ peaks, even for bulk water, remains controversial in modern literature, due to the complexity of the H-bond network of water (presence/absence of up to two donors and two acceptors for each water molecule) and the symmetry of the $\nu_{OH}$ vibration (symmetric vs. antisymmetric)[54–56]. Here, we do not aim to precisely attribute each $\nu_{OH}$ peak to a specific H-bond configuration and/or vibrational symmetry, which would be practically unrealistic. Rather, we assign each peak to an approximate corresponding H-bond number (tetrahedral, trihedral, etc.) following existing interfacial water studies[17,19,57,58]. Within one broad $\nu_{OH}$ spectrum, the typical consensus is that peak components with higher wavenumbers roughly correspond to lower H-bond coordination numbers[17,19,57,58].

Following the above approach, we attribute OH-1 to an H-bond number close to 4 (either four-fold or a combination of four-fold and three-fold coordination), and OH-2 to an H-bond number near 3 (either three-fold or a combination of three-fold and two-fold coordination)[19,57–61]. The assignments for OH-3 to OH-5 will be discussed later. In comparison, bulk aqueous solution only has two $\nu_{OH}$ peak components, OH-1 and OH-2 (Supplementary Fig. 19). Compared to the bulk solution, the pristine HOPG–solution interface features a much smaller fraction of OH-1, slightly larger OH-2, and the emergence of OH-3 to OH-5, indicating strong H-bond breaking effects. After ageing, the overall H-bond number of water at the HOPG–solution interface became higher, as evidenced by the increased area fractions of OH-1 and OH-2 and decreased fractions or absence of OH-3 to OH-5. This reveals that the hydrocarbon layers are less effective in breaking H-bonds of nearby water compared to HOPG. The red shift of the peaks after ageing, on the other hand, is likely due to the weaker hydrophobic interaction between water molecules and the dynamically moving, less dense hydrocarbon layers (aged interface) compared to the fixed, densely packed HOPG substrate (pristine interface). In addition, adventitious hydrocarbons contain C–H bonds with small dipole moments, which can enable weak dipolar interactions with the surrounding water[62], thus mitigating H-bond disruption effects.

In SHINERS measurements, the active interfacial liquid regions contributing to the signal are sandwiched between the HOPG surface and SiO$_2$ (shell of the Au/SiO$_2$ particles) (Fig. 1d). To rule out the possible role of silica in producing the emergent interfacial $\nu_{OH}$ peaks (OH-3 to OH-5), we designed and fabricated a control sample, Au film/SiO$_2$/electrolyte/SiO$_2$/Au nanoparticles. As shown in Supplementary Fig. 21, the silica layer is ≈3 nm thick at the active gap region. This sample serves as a well-defined control system that enables Raman enhancement, is graphite-free, and ensures SiO$_2$ is the dominant surface composition in contact with the probed volume of aqueous solution. SHINERS spectra of this control sample at multiple different areas and varying electrode potentials did not exhibit OH-3 to OH-5 peaks (Supplementary Fig. 21), distinct from the results obtained for HOPG/solution interfaces. These data prove that neither the silica shell layer nor the electrical double layer formed on the shell contributed measurably to the SHINERS peaks of the HOPG–water interface.

Combining 3D-AFM and SHINERS results, we propose the overall structure of the HOPG–water interface at OCP, as summarized in Fig. 2g. A clean, as-prepared interface is in state 1, featuring multiple (at

least 2–3) water layers separated by ≈3 Å. The interfacial water has strongly broken H-bonds compared to bulk water and a diverse distribution of H-bond configurations. After ageing, the interface evolves into state 2, which is dominated by 2–3 layers of hydrocarbon molecules with 4–5 Å separation. Water is mostly depleted from the hydrocarbon layers. The water molecules above the hydrocarbons have overall higher H-bond coordination numbers than those at pristine HOPG surfaces, yet still lower than that of bulk water.

## Electrified graphite–water interfaces: pristine response

After identifying the two distinct states of the HOPG–water interfacial structure, we proceeded to investigate their potential dependence. As is common for graphite and metallic electrodes, positive potentials tend to induce surface oxidation[19,63,64]. Therefore, to preserve the electrode structure and to enable the generalization to a broad range of electrode–water interface systems, we applied negative potentials to the HOPG electrode. The potential range was typically ≈0 to −2.2 V vs. Ag/AgCl, large enough to examine rich reconfiguration effects of the interfacial structure. At sufficiently negative potentials, the hydrogen evolution reaction (HER) became significant (Supplementary Fig. 22), producing bubbles in the local area that prevented further 3D-AFM and SHINERS measurements. The bubble onset potential varied for different samples and local areas, but it remained in the −1.6 to −2.6 V range (vs. Ag/AgCl).

In previous microscopy or spectroscopy studies of electrified solid–water interfaces, there was a lack of control of the initial interfacial configuration at OCP, as the system could be in either state 1 or state 2, or an intermediate regime[12,18–20,28,65]. This is likely the key reason why existing literature shows persistent discrepancies regarding the effect of electrode potential/electric field on the interfacial water configuration (Supplementary Tables 1–4). To resolve such controversies, here we first prepared the HOPG–aqueous solution system to state 1, before examining the potential dependence. State 1 was achieved by either limiting the total sample preparation/air exposure time to no more than a few minutes before starting the measurement, or via electrochemical cleaning after the sample was prepared (as discussed later). State 1 was confirmed when $d_{12}$ was observed to be ≈3 Å in 3D-AFM, or when FP and $v_{CH}$ peaks were either negligible or much weaker than the $v_{OH}$ peak in SHINERS.

We first performed potential-dependent 3D-AFM in a series of aqueous solutions containing either $K_2SO_4$ (0.01–0.2 M) or KCl (0.1 M). A total set of 9 independent 3D-AFM measurements were conducted with a starting condition of $d_{12}$≈3 Å (state 1) at OCP, as shown in Fig. 3a, b, Supplementary Figs. 5–11. To minimize the possible convolution of ageing to the potential-dependence of the interfacial water structure, the measurements were conducted either quickly (within ≈20 min for each potential scan) to mitigate ageing or in an argon-sealed liquid cell to slow down the ageing process. As shown in Supplementary Figs. 5–10, argon sealing was usually effective in preserving the pristine interface for at least 30 h. All of the obtained force curves, regardless of the measurement location (Urbana or Madrid) and environmental condition, revealed that $d_{12}$ is remarkably potential independent within at least 0 to −2.2 V vs. Ag/AgCl. While each individual set of 3D-AFM results showed small variations of $d_{12}$, mostly within 3.0–3.5 Å, the overall average values were always near 3.3 Å at each of the applied potentials (Fig. 3b).

The observed potential independence of interlayer distance might seem counterintuitive, since potential modulations are typically expected to induce changes in the ionic concentration at the interface. However, as discussed earlier, for dilute aqueous solutions (<1 M) at solid interfaces, both our results and a large body of prior publications supported the salt concentration-independence of the ≈3 Å hydration layer spacing[9,11,30,41,42]. Therefore, as long as water is still the dominant species within the

first 1–2 nm from the HOPG surface in the potential range we studied, it is not surprising to observe a constant ≈3 Å interlayer distance.

Although the interlayer spacing is potential-independent, the polar water molecules at the interface are expected to change their orientations and thus the H-bond configurations at different electric fields. We investigated such effects through multiple sets of SHINERS measurements (Fig. 3c, Supplementary Figs. 23, 24). Similar to 3D-AFM measurements, we first prepared the samples to reach state 1 at OCP, as evident from the weak or negligible FP and $v_{CH}$ peaks. As shown in Fig. 3c, hydrocarbon-related modes were absent not only at OCP but also throughout all the applied potentials, confirming that the interface remained pristine during the whole measurement process. The $v_{OH}$ mode consistently exhibited five peak components across all applied potentials. At more negative potentials, the overall $v_{OH}$ band became broader while the height decreased. The total area of $v_{OH}$ remained nearly constant (Fig. 3d), again confirming that the total amount of interfacial water molecules was potential-independent within the sensitivity limit. This is consistent with the potential-independent force curves and $d_{12}$ in the 3D-AFM results (Fig. 3a, b). Since water molecules are neutral yet polar, an interfacial electric field can induce the reorientation of the molecules without changing their average distance from the electrode surface, thus preserving the $d_{12}$ and average interfacial density.

We further analyzed the potential-dependence of the individual $v_{OH}$ peaks (OH-1 to OH-5). OH-1 to OH-4 exhibited red-shifts at more negative potentials (Fig. 3e), likely due to Stark effects. In contrast, OH-5 remained at the same position (≈3700 cm⁻¹) at different potentials. The width of all the peaks exhibited certain levels of increase at more negative potentials (Fig. 3f), leading to an overall broadening of the $v_{OH}$ band. Among the five peaks, OH-2 showed the most pronounced increase in peak width. The broadening effects are likely due to the larger variations in the H-bond configurations that contribute to each peak, as a result of the interfacial electric field. For example, the OH-2 peak may consist of at least two configurations, one with two donors and one acceptor (DDA), the other with one donor and two acceptors (DAA)[17,59]. At a stronger electric field, the difference in the vibrational energy of these two configurations may become larger; in addition, each configuration may be further split into sub-configurations with different vibrational energies resulting from their orientational differences (e.g., parallel vs. vertical to the electric field). These effects can lead to a broadening of the OH-2 peak. We further observed that the area fraction of OH-2 decreased, OH-3 was nearly constant, and OH-1, OH-4, and OH-5 increased at more negative potentials (Fig. 3g, h). The evolution in area fractions also signifies the diversification of H-bond configurations, where some OH-2 structures split into configurations with either larger (OH-1) or smaller (OH-3 to OH-5) H-bond numbers, due to the orientation-dependent free-energy modulations induced by the interfacial electric field.

The higher-wavenumber peaks, OH-3 to OH-5, are expected to correspond to configurations with low H-bond numbers, such as 2-coordinated water, cation-coordinated water, dangling O–H, and nondonor (ND) water, although the exact peak assignments remain highly controversial[17,19,57–61]. Nevertheless, one special property worth noting is the lack of Stark shift of OH-5 over the large range of potentials, which was consistently observed in each of the 7 potential scans over 3 independent sets of measurements where OH-5 was present (Fig. 3c, e, Supplementary Figs. 23–26). Stark shift at a given electric field, to the first-order approximation, is proportional to the difference in the dipole moment between the ground and excited state along the direction of the electric field[66,67]. Therefore, the OH-5 configuration likely has a nearly fixed molecular orientation (no significant fluctuation over time) with a difference dipole parallel to the substrate surface (perpendicular to the interfacial electric field), thus having zero Stark effect. Among all the reported weakly coordinated H-bond

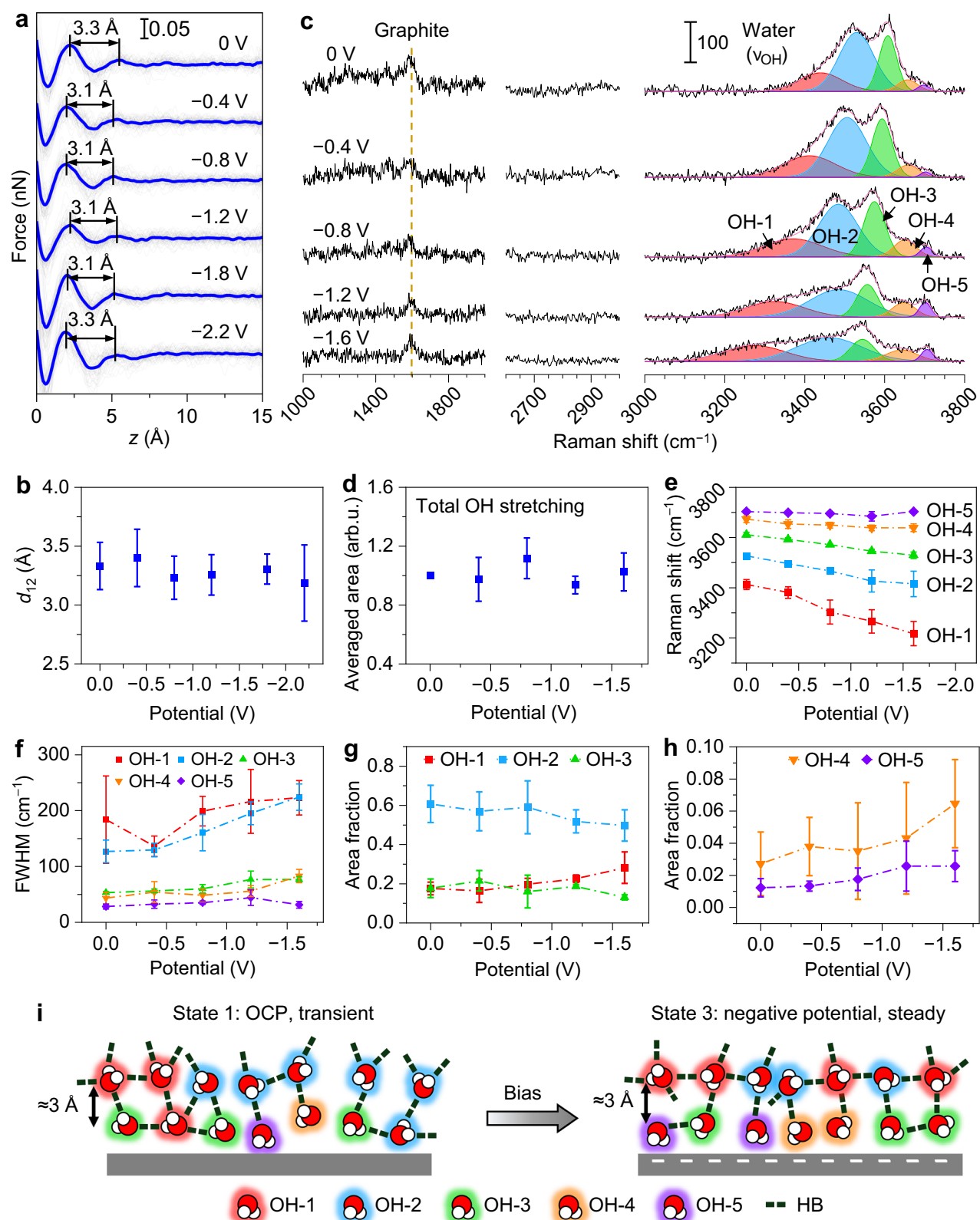

configurations, there is only one possible state fulfilling this criterion, the antisymmetric OH stretching of ND water. The ND configuration, proposed before in a previous study of Au–water interface[18], may exist at the first hydration layer with oxygen pointing up (away from the substrate) and two hydrogen atoms down (in close proximity to the substrate). While the ground state dipole moment is vertical, the difference dipole of the antisymmetric OH stretching is parallel to the

substrate, resulting in zero Stark shift. We thus assign OH-5 to the antisymmetric $\nu_{OH}$ of ND water.

Figure 3i illustrates the overall potential-dependent response of the pristine HOPG–water interface, combining results from 3D-AFM and SHINERS. At negative potentials, the originally transient state 1 evolves into a steady state 3. In this process, the interface remains water-dominated, with a constant ≈3 Å hydration layer spacing and the

**Fig. 3 | Effect of electrode potential on pristine HOPG/aqueous solution interfaces. a** Force–distance curves extracted from 3D-AFM maps of a pristine HOPG/0.01 M $K_2SO_4$ aqueous solution interface at different potentials vs. Ag/AgCl (Urbana). Gray: average curves of single $x$–$z$ maps, blue: average of the gray curves. $d_{12}$ is marked on each curve. **b** Average values of $d_{12}$ as a function of the electrode potential. Statistics were derived by pooling technical replicates from 6 experimental runs (including the ones depicted in (**a**) and Supplementary Figs. 5–10). The precise sample sizes ($n$) are $n = 24$ (0 V), $n = 20$ (−0.4 V), $n = 22$ (−0.8 V), $n = 13$ (−1.2 V), $n = 9$ (−1.6 V) and $n = 4$ (−2.2 V). **c** SHINERS spectra of a pristine HOPG/0.1 M $K_2SO_4$ aqueous solution interface at a series of electrode potentials vs. Ag/AgCl (Urbana). The $\nu_{OH}$ mode is deconvoluted into 5 Voigt peaks, OH-1 to OH-5 (red, blue, green, orange and violet, respectively). The total fit is shown in pink. **d**–**h** SHINERS metrics of pristine HOPG/0.1 M $K_2SO_4$ solution interface as a function

of electrode potential ($n = 3$ pooled from 2 experimental replicates). Data were extracted from SHINERS results in (**c**) and Supplementary Figs. 23, 24. The metrics include **d** the averaged total area of the $\nu_{OH}$ peak (before averaging, peak areas were normalized by scaling the area of the 0 V data to 1 for each potential scan), **e** Raman shift (peak position) of OH-1 to OH-5, **f** full width at half maximum (FWHM) of OH-1 to OH-5, **g** area fractions of OH-1 to OH-3 (ratio of each peak versus the total area of the $\nu_{OH}$ mode), and **h** area fraction of OH-4 and OH-5. All statistics are presented as mean ± standard deviation (SD). **i** Schematic representation of the potential-driven transition of a pristine graphite–water interface from state 1 to state 3, illustrating the characteristic observations: constant $d_{12}$, constant amounts of interfacial water, and H-bond diversification (less OH-2 and more OH-1, OH-4, and OH-5) at negative potentials. Source data are provided as a Source Data file.

---

same overall interfacial water density. The H-bond configurations of the interfacial water become more diverse, featuring a simultaneous increase of both the ice-like 4-coordinated structures and a few low-coordinated states, including the monomer-like or gas-like ND water.

### Electrified graphite–water interfaces: non-pristine response

We further investigated the influence of negative potentials on the initially non-pristine HOPG–water interface. State 2, featuring $d_{12} \approx 4$–5 Å or FP (or $\nu_{CH}$) peak intensity higher than that of $\nu_{OH}$, was reached either unintentionally right after sample preparation (due to adventitious airborne or liquid-borne hydrocarbons), or intentionally through >1 h ageing under ambient conditions after the HOPG surface was immersed in the aqueous solution.

Starting from state 2, we conducted a series of 6 potential-dependent 3D-AFM experiments (Urbana and Madrid), with $K_2SO_4$ in water (0.01–0.2 M) and KCl in water (0.1 M) as the electrolyte (Fig. 4a, b, Supplementary Figs. 12, 13). All results consistently showed an abrupt decrease of $d_{12}$ from 4–5 Å to ≈3 Å at negative potentials, where the transition mostly occurred between −1 to −1.5 V vs. Ag/AgCl. After the transition, further decrease of the electrode potential did not induce more changes in $d_{12}$, which remained at ≈3 Å in all the data sets (Fig. 4a, b, Supplementary Figs. 12, 13). These results suggest that hydrocarbons were removed from the interface and replaced by water molecules at sufficiently negative potentials, and the water-dominated interfacial configuration was stable under the negative polarization conditions.

Right after the potential was changed back to OCP, the force curves remained mostly unchanged, with the same ≈3 Å interlayer distance (Fig. 4a, b, Supplementary Figs. 6, 7, 13a, 14). However, after about 1 to 2 h under OCP condition, with the aqueous solution exposed to ambient air, we observed a transition of $d_{12}$ back to ≈4–5 Å (Supplementary Fig. 14). These results are consistent with the ageing effects shown in Fig. 2, confirming that the system reached the transient state 1 right after the potential was switched from negative to OCP, before evolving into the steady state 2 over time under OCP.

We also conducted a series of potential-dependent SHINERS measurements for HOPG/0.1 M $K_2SO_4$ in water starting from state 2, with results summarized in Fig. 4c, Supplementary Figs. 25–27. In all these results, FP and $\nu_{CH}$ bands were much stronger than that of the $\nu_{OH}$ at initial OCP or 0 V (vs. Ag/AgCl) condition, consistent with state 2. At negative potentials, the hydrocarbon FP and $\nu_{CH}$ bands became weaker and eventually negligible at ≈−1 to −2 V vs. Ag/AgCl (Fig. 4c, d, Supplementary Figs. 25–27). In contrast, the overall intensity of $\nu_{OH}$ became stronger. After the potential was gradually changed back to 0 V (ramping from −2 to 0 V in ≈1 h), the hydrocarbon-related peaks only slightly recovered, while the overall intensity/area of $\nu_{OH}$ was largely preserved (Fig. 4c, d, Supplementary Figs. 25, 26). These results further confirmed the transition from hydrocarbon to water at the HOPG–aqueous solution interface at negative potentials (state 2 to state 3), as well as the reversal to the pristine state 1 immediately after the removal of the electrode potential.

After decomposing the $\nu_{OH}$ band, we identified up to two peaks, OH-1 and OH-2, within the ≈0 to −1.2 V (vs. Ag/AgCl) potential range (Fig. 4c, Supplementary Figs. 25b, 26b, 27b). In this regime, at more negative potentials, the two peaks red-shifted due to the Stark effect and became wider (Fig. 4e, f). These observations are consistent with the potential-induced $\nu_{OH}$ peak component evolutions of the initially pristine interface (Fig. 3c–h), revealing that the water surrounding the hydrocarbon layers also exhibits H-bond diversification effects with increasing interfacial electric field. We further observed that the area fractions of OH-1 and OH-2 remained mostly constant (Fig. 4g, 0 to −1.2 V), indicating that the average H-bond number remained stable within this potential range. As the potential reached −2 V, the OH-3 to OH-5 components suddenly emerged, at the same time when the hydrocarbon-related peaks nearly completely disappeared (Fig. 4c). This transition, together with the ageing effect shown in Fig. 2c, reveals that direct contact between water and the graphite surface is a prerequisite for the formation of the low-coordination OH-3 to OH-5 configurations. The interfacial electric field, on the other hand, only modulates the dispersion within the existing modes without introducing emergent H-bond configurations. Additionally, at −2 V, the peak position, FWHM, and area fractions of OH-1 to OH-5 (Fig. 4e–g) are mostly consistent with those of the initially pristine system at high negative potentials (Fig. 3e–h), further confirming that the same pristine state 3 was reached in both cases.

As the potential returned to 0 V, nearly all the features of OH-1 to OH-5 were restored to those characteristics of the pristine state 1 (compare the final 0 V data in Fig. 4e–g to the initial 0 V data in Fig. 3e–h). Note that small hydrocarbon FP and $\nu_{CH}$ bands emerged at the final 0 V state (Fig. 4c). This was due to the gradual change of potential from −2 V to 0 V (see Supplementary Fig. 25c, d) that took ≈1 h, during which small amounts of hydrocarbons accumulated at the interface. Nevertheless, the interface was still dominated by water at the final 0 V state, and the $\nu_{OH}$ modes remained similar to those of the pristine states. Additionally, the change in the initial and final 0 V spectra in Fig. 4c cannot be explained by other potential-induced hysteresis effects besides the change of hydrocarbons, as the pristine, water-dominated interface did not exhibit any hysteresis effect (Supplementary Fig. 23).

As another indicator of the interfacial liquid structure, we recorded the current transients during SHINERS measurements (Supplementary Fig. 28). The water-dominated interface exhibited larger transient currents than those of the hydrocarbon-dominated one, due to the expected higher double-layer capacitance of the pristine interface.

The overall structural transitions among all three observed states of the HOPG–water interface are summarized in Fig. 4h. Despite the distinct interfacial structures in states 1 and 2, they converge to the same state 3 under sufficiently negative electrode potentials. We propose two possible origins of this effect. One is the change of the interfacial free energy. Since water is polar while hydrocarbons are nonpolar, an interfacial electric field will lower the enthalpy of water

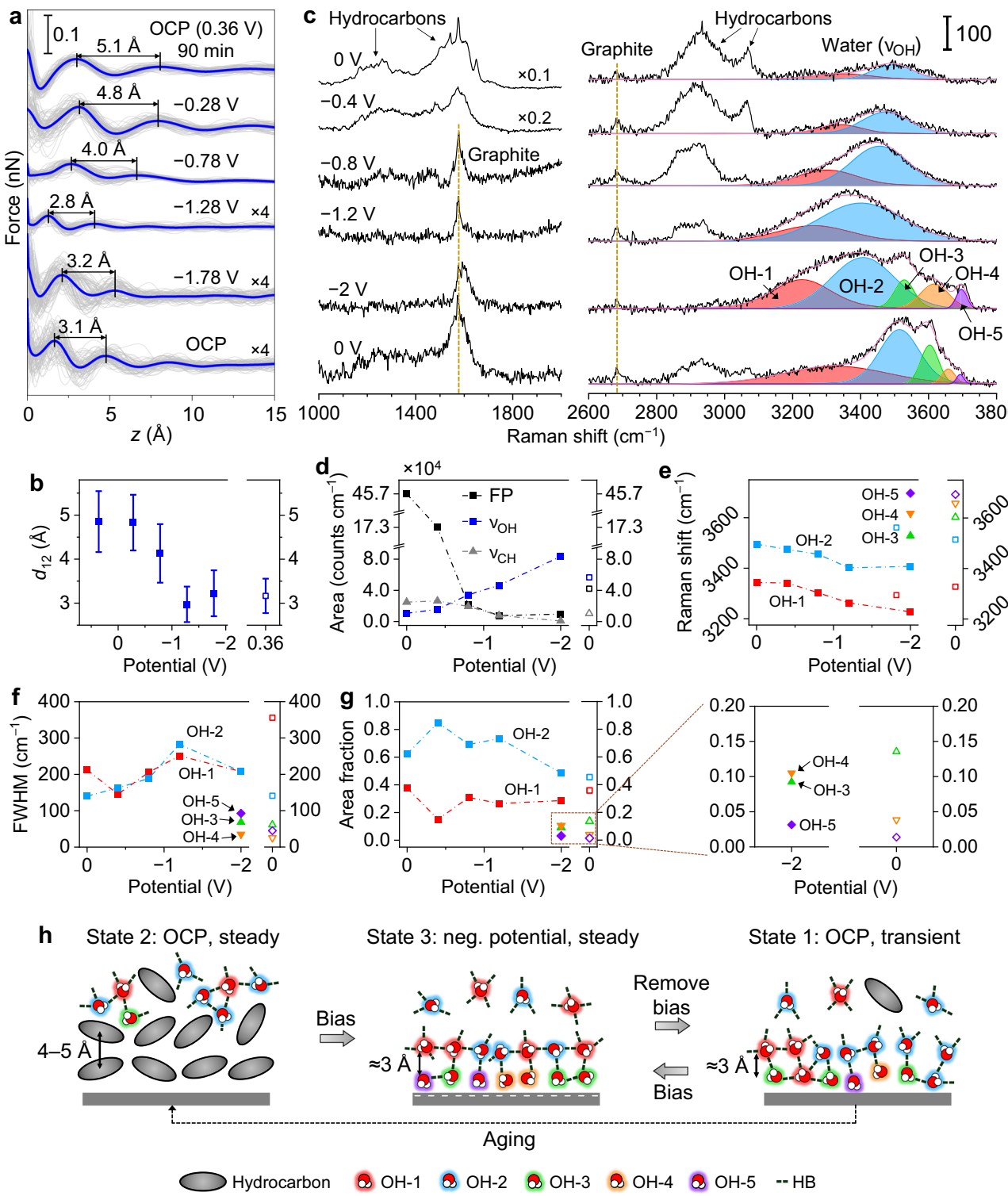

**Fig. 4 | Influence of the electrode potential on non-pristine HOPG/aqueous solution interfaces. a** Force−distance curves extracted from 3D-AFM maps of an initially non-pristine HOPG/0.1 M KCl in water interface at a series of electrode potentials (Madrid). Gray: individual force−distance curves; blue: averaged curve. $d_{12}$ is marked on each curve. Before starting the measurements, the HOPG surface was immersed in the 0.1 M KCl solution surrounded by ambient air for 90 min to reach a non-pristine state (state 2). The sequence of electrode potentials proceeded as OCP (0.36 V), −0.28 V, −0.78 V, −1.28 V, −1.78 V (vs. Ag/AgCl) and OCP. **b** $d_{12}$ extracted from (**a**) and plotted as a function of the electrode potential. Data represented as mean ± SD ($n$ = 80 technical replicates). **c** SHINERS spectra of a non-pristine HOPG/0.1 M $K_2SO_4$ in water interface at different electrode potentials (Urbana). Before measurements, the electrochemical cell was exposed to ambient

air for ≈270 min to reach a non-pristine state (state 2). The sequence of potentials proceeded as 0 V, −0.4 V, −0.8 V, −1.2 V, −2 V and 0 V (vs. Ag/AgCl). The $\nu_{OH}$ mode is resolved into 5 Voigt profiles, labeled OH-1 through OH-5 (represented by red, blue, green, orange and violet curves). The total fit (pink line) represents the sum of these individual peaks. **d**–**g** SHINERS metrics of the initially non-pristine HOPG/0.1 M $K_2SO_4$ solution interface as a function of electrode potential. Data were extracted from the results in (**c**). The obtained metrics include **d** area under the FP, $\nu_{CH}$ and $\nu_{OH}$ peaks, and **e** Raman shift (peak position), **f** FWHM and **g** area fraction (ratio of each peak vs. the total area of the $\nu_{OH}$ mode) for OH-1 to OH-5. **h** Schematic representations of the overall HOPG−water interfacial structural transitions due to ageing and electrode potential. Source data are provided as a Source Data file.

molecules with the alignment of the dipoles to the field, while the hydrocarbons' free energy is not affected. When the electrostatic free energy modulation of the water dipoles is sufficiently large, it may overcome the interaction energy between the hydrocarbons and the graphite surface, resulting in the swapping between water and hydrocarbons at the interface. Simplified calculations show that this scenario is plausible (Supplementary Note 2). The other possibility is electrochemical reactions, including hydrogen evolution and direct hydrocarbon reduction. Cyclic voltammetry measurements revealed hydrogen evolution in the measured potential range (Supplementary Fig. 22), which can produce hydrogen bubbles that may propel the hydrocarbons away from the electrode surface. Additionally, direct electroreduction of hydrocarbons which contain C–O, C–F and aromatic rings is also possible[68–70].

To conclude, we have resolved the long-standing controversy on the structure of graphite–water interfaces. By combining microscopy and spectroscopy methods, and precisely controlling the initial sample preparation, ageing time, and electrode potential, we identified the hitherto hidden co-existence of three interfacial states. At an open circuit or a nearly neutral surface, the interface can be in either the pristine, water-dominant state or the non-pristine, hydrocarbon-dominant state. At sufficiently negative potentials, the interface inevitably transitions into a pristine state with a diverse distribution of hydrogen bond configurations, including monomer-like non-donor water. The hydrogen bond configurations at the interface are subject to modulations of three factors: the graphite surface, the hydrocarbon layer, and the interfacial electric field. We have distinguished the contribution of each factor and constructed a universal interfacial configuration diagram under varying surface charge and ageing conditions.

## Implications

We expect the results to be applicable to not only graphite, but also a large range of other solid–water interfaces. Except for a few model systems, such as freshly cleaved mica or calcite, most real-life and industrially relevant solids (e.g., most semiconductors and metals) exhibit certain levels of hydrophobicity[71–73]. Oftentimes, these surfaces become more hydrophobic upon air exposure, due to the accumulation of hydrocarbons or other airborne species[71–73]. We anticipate that these solid–water interfaces will also exhibit multiple interfacial states, which are subject to ageing in ambient environments and to electric field-driven transitions from non-pristine to pristine states, accompanied by strong hydrogen bond reconfigurations of the interfacial water.

The ability to identify the realistic structure of solid–water interfaces and deconvolute the contributing factors will be crucial for the rational design of relevant systems for practical applications, such as electrochemical energy conversion and storage, biosensing, and photocatalysis.

## Methods
### Materials
Highly oriented pyrolytic graphite (HOPG) with a mosaic spread of 0.8° ± 0.2° (ZYB grade) was obtained from Bruker Corporation.

Madrid: Potassium sulfate (anhydrous, >99%, Sigma-Aldrich) or potassium chloride (≥99.0%, Sigma-Aldrich) and ASTM D1193 Type I deionized (DI) water (resistivity of 18.2 MΩ cm, ELGA Maxima) were used for preparing electrolyte solutions of various concentrations. Glass vials were rinsed with acetone (≥97.0%, Sigma-Aldrich), isopropanol (99.6%, Acros Organics) and DI water before preparing the electrolyte solutions.

Urbana: Potassium sulfate (anhydrous, >99%, Acros Organics) and ASTM D1193 Type I DI water (resistivity of 18.2 MΩ cm; Synergy UV system, Millipore Sigma) were used for preparing electrolyte solutions of various concentrations. Glass vials were rinsed with acetone,

isopropanol, and DI water before being used to prepare electrolyte solutions.

### 3D-AFM
3D-AFM measurements were independently performed in Madrid (Spain) and Urbana (IL, USA) using amplitude modulation mode with similar methodologies. In both cases, the cantilever was photo-thermally excited at its first resonance frequency. A low-frequency sinusoidal signal (10–100 Hz) was applied to the $z$-piezo to modulate the tip–sample distance, synchronized with the $x$-displacement, to ensure a full $z$-cycle at each $x$-position. Amplitude and phase signals were recorded as functions of $x$, $y$ and $z$. We found that the signal obtained over $x$–$z$ frames and 3D volumes tends to better represent the overall interfacial structure compared to a series of force–distance curves obtained at a fixed $(x, y)$ point. The latter approach is sensitive to the local variations of the substrate surface, such as the lattice sites, impurities, adventitious adsorbates, etc., and is thus less representative of the overall interfacial liquid configuration.

In Madrid, measurements were performed using a Cypher S system (Asylum Research Inc., USA) with custom-developed 3D-AFM software. In Urbana, a Cypher ES system (Asylum Research Inc., USA) with the AC Fast Force Mapping (AC FFM) module was used. Silicon probes were used for the 3D-AFM measurements: Arrow-UHFAuD (NanoAndMore, Germany) in Madrid, and FS-1500AuD (Asylum Research) and PPP-NCHAuD (Nanosensors) in Urbana. Cantilever calibration was performed using the contactless Sader method, as implemented in the commercial AFM software as GetReal.

Cantilevers were cleaned using sequential solvent-rinsing followed by UV-ozone treatment. In Madrid, cantilevers were immersed in a mixture (50:50 in volume) of isopropanol (99.6%, Acros Organics) and DI water, then thoroughly rinsed with DI water. In Urbana, cantilevers were sequentially soaked in acetone (≈30 min), isopropanol (≈12 h) and DI water (≈1 h), then dried using argon (6 N, Airgas) or a compressed air duster (Office Depot). After drying, UV-ozone treatment was applied. In Madrid, cantilevers were exposed for ≈1 h using a PSD-UV3 system (Novascan Technologies, USA). In Urbana, ≈5 min exposure was performed using a ProCleaner Plus system (BioForce Nanosciences), with the AFM probe positioned ≈5 cm from the UV source.

Additional details on the cantilever calibration values and 3D-AFM parameters are summarized in Supplementary Tables 5–8.

Electrode potential control was implemented using either a three-electrode or a two-electrode configuration (see Supplementary Fig. 29)[16,74–76]. In both lab setups, freshly cleaved HOPG was used as the working electrode and platinum served as both a quasi-reference and a counter electrode. In the Madrid lab, a three-electrode setup was used with platinum foils (0.1 mm thick, >99.99%, Sigma-Aldrich) connected to a PalmSens4 potentiostat (PalmSens). The platinum foils were cleaned using piranha solution (2:1 $H_2SO_4$:$H_2O_2$), followed by sonication in DI water for 5 min. In the Urbana lab, a two-electrode setup was employed, using a platinum ring as the combined counter and quasi-reference electrode. The ring was cleaned by sequential rinsing in acetone, isopropanol and DI water, followed by drying with argon or a compressed air duster. Electrode potential was applied and monitored using a Keithley 2450 source meter. Urbana's Cypher ES featured environmental control of the cell chamber, which was purged with argon for ≈1–2 min and tightly sealed, enabling stable operation over at least 2 days without any noticeable change in liquid volume.

All measurements were performed at room temperature and ambient pressure.

### 3D-AFM data processing
Established force reconstruction methods[77,78] were implemented using custom-written codes to reconstruct the conservative component of the AFM probe–sample interaction force. To enhance the force signal

and improve the visualization of interfacial layers, the Madrid lab applied binomial smoothing (range = 2) followed by background subtraction using a single-exponential fit, while the Urbana lab employed a 50-point moving average and a double-exponential background subtraction.

In the Madrid lab's workflow, a single representative $x$–$z$ map was selected from each 3D-AFM data set and analyzed in detail by examining the individual force–distance curves and their average. The substrate position ($z = 0$) was defined based on the setpoint amplitude. The Urbana lab averaged each $x$–$z$ map within a 3D-AFM set into a single force–distance curve. Prior to averaging, the force–distance curves were aligned based on the tip–sample separation corresponding to a phase shift of 22° below the free oscillation phase. The substrate position ($z = 0$) was then defined as the $z$-position (below the first water layer) where the force value equaled that at the first water layer position, following protocols in ref. 6. The $x$–$z$ map-averaged force–distance curves were then collectively analyzed, and their overall histogram and average were used to represent the interfacial structure of the entire 3D volume.

### Raman spectroscopy

Raman spectroscopy experiments were performed at Urbana. All Raman measurements were carried out using a confocal Raman imaging system (Horiba, LabRAM HR 3D-capable Raman spectroscopy) with a 633 nm excitation laser at 3.5 mW power. The beam was focused onto the sample surface using either a 50 × (with numerical aperture (NA) of 0.5) or 10 × (NA 0.25) objective. Scattered light passed through a 300 grooves per mm grating and was directed to a Horiba Synapse back-illuminated deep-depletion charge-coupled device (CCD) camera. Raman spectra from 300 to 4000 cm$^{-1}$ were collected with an accumulation time of 20–30 s, and repeated 3–6 times for averaging unless otherwise specified. A silicon wafer was used for spectral calibration at the beginning or end of the experiment.

A Raman spectrum of bulk 0.1 M $K_2SO_4$ aqueous solution (Supplementary Fig. 19) was obtained with the laser focused on a HOPG surface in the absence of $Au/SiO_2$ core/shell nanoparticles. A 50 × objective (NA 0.5) and an acquisition time of 20 s with 6 accumulations were used.

The preparation of $Au/SiO_2$ core/shell particles was described elsewhere[45] and is briefly explained here. A 10 mL 0.01 wt.% $HAuCl_4$ (≥99.9 %, Sigma Aldrich) aqueous solution was first heated to boiling, then 0.07 mL of 1 wt.% sodium citrate aqueous solution (>99%, Alfa Aesar) was added to the boiling solution and kept heated for 30 min. The resulting solution was cooled to room temperature and characterized by a UV-vis spectrophotometer. A characteristic absorption peak at ≈534 nm indicated the formation of ≈60 nm Au nanoparticles (Supplementary Fig. 17a). For silica coating, 3 mL of the synthesized bare Au particles solution, together with 0.04 mL 1 mM APTMS solution (Sigma Aldrich) and 0.32 mL 0.54 wt.% sodium silicate solution (Sigma Aldrich) was heated to 90 °C and kept for 2 h. The solution was then cooled in an ice bath. To remove ligands and other by-products, 3 mL of the as-synthesized particles were centrifuged at 2020 × $g$, 10 min. After discarding the supernatant, the particles were re-dispersed in 1.5–3 mL of DI water. This centrifugation and supernatant removal process was repeated up to five times, and the final product was dispersed in 200 μL of DI water. Transmission electron microscopy (TEM) verified the morphology of the $Au/SiO_2$ particles, while cyclic voltammetry confirmed that the silica shell is pinhole-free (Supplementary Fig. 17b, c).

SHINERS measurements were performed using either a commercial EC cell purchased from DEK Research Instrumentation (Supplementary Fig. 29d) or a similar home-made cell. The as-prepared $Au/SiO_2$ core/shell particle solution was drop-casted onto the freshly cleaved HOPG surface and dried in air for 1–2 h. During drying, the surface usually accumulated organic contaminants, which were subsequently removed by negative electrode polarization (electrochemical cleaning). Further, drying often resulted in non-uniform particle distribution, forming coffee ring-like structures. SHINERS spectra were typically collected ≈20–30 μm from the coffee ring's edge. A sub-monolayer coverage is expected in these areas (Supplementary Fig. 18). An Ag/AgCl reference electrode (BaSi, 3 M NaCl, MF-2052) and a Pt wire counter electrode (CH Instrument, CHI115) were used in the SHINERS measurements.

All measurements were performed at room temperature and ambient pressure.

The collected Raman spectra were processed as follows: cosmic spikes (sharp peaks) were first removed, followed by baseline correction using a manual spline fit. For the OH-stretching region, the bulk spectrum was subtracted (see Supplementary Figs. 19, 20). The resulting OH-stretching spectra were then analyzed using Voigt profile fitting.

### Electrode potential calibration

All the results in this work are reported against Ag/AgCl, unless otherwise specified. The potential difference between the Pt quasi-reference and a standard Ag/AgCl reference electrode (BaSi, 3 M NaCl, MF-2052) was obtained by comparing the corresponding equilibrium redox potential of the redox couple Ferri/Ferrocyanide (5 mM) in 0.1 M $K_2SO_4$ solution. 0 V against Pt was found to be equal to 0.22 V vs. Ag/AgCl (Supplementary Fig. 30). It is worth noting that the potential of a Pt quasi-reference electrode may not be exactly fixed; variations in the rough range of ±0.1 V may arise but do not affect the conclusions we reached in this work.

Additionally, electrochemical impedance spectroscopy (EIS) measurements of HOPG in 0.1 M $K_2SO_4$ solution revealed a minimum differential capacitance at −0.6 V vs. Pt (−0.38 V vs. Ag/AgCl) (Supplementary Fig. 31). While this value may be close to the potential of zero charge (PZC), the exact PZC value may change at different interfacial states, depending on the presence/absence of adventitious hydrocarbons.

### Reporting summary

Further information on research design is available in the Nature Portfolio Reporting Summary linked to this article.

## Data availability

The data that support the findings of this study are available from the corresponding authors upon request. Unprocessed spectroscopic data are provided as Supplementary Data 1. Source data are provided with this paper.

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

## Acknowledgments

L.K.S.B., F.Z., J.K., Q.A., S.Z., K.S.P., and Y.Z. acknowledge support from the National Science Foundation under Grant No. 2339175, the Beckman Young Investigator Award provided by the Arnold and Mabel Beckman Foundation, and the Sloan Research Fellowship from the Alfred P. Sloan Foundation. L.K.S.B. acknowledges support from the TechnipFMC Educational Fund Fellowship. J.K. was partially supported by a PPG-MRL Graduate Research Assistantship. Q.A. acknowledges support from the PPG-MRL Graduate Research Assistantship program. R.G. acknowledges financial support from Ministerio de Ciencia e Innovación grants PID2022-136851NB-I00/ AEI/10.13039/501100011033 and EUR2022-134029, as well as the European Commission Horizon Europe MSCA Doctoral Network NANORAM, Grant No. 101120146. R.G. also acknowledges Zhen Tang for helping to process some AFM images.

## Author contributions

Y.Z. designed the experiments in Urbana. R.G. designed the experiments in Madrid. L.K.S.B., D.M.A., F.Z., Q.A., S.Z., and K.S.P. conducted 3D-AFM experiments and initial analyses. F.Z. and J.K. synthesized Au/SiO$_2$ nanoparticles and performed SHINERS measurements. L.K.S.B. and F.Z. performed electroanalytical measurements. L.K.S.B., D.M.A., and F.Z. conducted in-depth data analyses. Y.Z. and R.G. supervised the work. L.K.S.B., F.Z., R.G., and Y.Z. wrote the manuscript with input from all co-authors.

## Competing interests
