## [Transparent Peer Review file · Nature Communications]

Probing the molecular structure at graphite–water interfaces by correlating 3D-AFM and SHINERS

Corresponding Author: Professor Yingjie Zhang

Version 0:

Reviewer comments:

Reviewer #1

(Remarks to the Author)

Re: Nature Communication Review25-49442-T “Correlative angstrom-scale microscopy and spectroscopy of graphite -water interfaces” by Bonagiri et al.

This is an excellent paper that combines AFM and SHINERS to investigate the hydration layers adjacent to HOPG surfaces in a variety of different, but nominally neutral pH, electrolytes. The core of the paper is aimed at resolving ambiguities associated with the observed spacing between hydration layers that have been observed to vary between ~ 0.3 nm and ~ 0.5 nm in different reports in the literature that include different or unspecified preparation conditions. By a combination of aging studies under open circuit conditions and polarization studies at negative potential the authors attribute the observation of thicker hydration layer to organic contamination that tends to build up with time under open circuit conditions. The authors show that this layer can be removed at negative potentials which is an interesting and helpful finding. Likewise, an important aspect of the work is that the contributions follow from interactions and parallel experiments performed between two independent laboratories that represent leading practitioners in fast force methods for probing the hydration layers at immersed interfaces. The effectiveness of sealing the cell is suggested by data in the supplement and perhaps more should be made of this in the main text. The authors provide Tables listing the imaging parameters in the supplement that will definitely help others interested in reproducing this work.

The paper is certainly worthy of publication. That said, the authors may wish to consider the following comments prior to going forward with publication.

1. In the case of the SHINERS experiments the preparation routine involved allowing the SHINERS particles to dry in the ambient for 1 to 2 hours. Did the surface become covered with organic contamination under these conditions? If not, does this say something about the difference between gas phase versus condensed phase exposure? Please clarify the history of the SHINERS experiments in this regard.
2. What is the prospect of intercalation reactions, e.g. proton or one of the other cations, during the excursion to negative potentials? Does the hysteresis between the water spectra e.g. the OH-1 opc water layers, Figure 2c, Figure 3c at 0 V versus that broadened and shifting OH-1 following the step to negative potentials in Figure 4C and returning to 0 V, speak to an irreversible component in the cleaning process. Could this be related to intercalation. The authors might wish to consider XPS analysis of the HOPG to see if the cation insertion is doing something to the HOPG under the conditions considered.After reading the supplement I see Figure 19 nicely addresses the hysteresis question and perhaps this should be mentioned in the main text.
3. Did you track the current transients associated with the potential step experiments? Is the conversion from the organic contaminated state charge dependent? Are the gas bubbles necessary to remove the organic contaminants whereby the gas/electrolyte interface surface area serves to scavenge that at the HOPG interface. Do the current transient for the potential steps reveals the presence of the contamination layers? After looking though the supplement I see the voltammograms...but the variation stated in the text are not evident..so the question remains...as to whether or not chronoamperometry of voltammetry can be used to probe for a clean surface.
4. In the introduction the statement is made that the hydration layer are independent of the identity of the supporting electrolyte even at concentrations in the range of 0.1 to 1 mol/L. This seems very unintuitive to me. In looking at the variation in d12 across the figure set, e.g. 0.29, 0.28, 0.33 nm in Figure 2, perhaps the claim of independence of d12 from ion identity is too strong a statement and rather the differences are yet to be revealed or understood due to sampling, analysis, entropic effects, etc. My sense is the variation evident in the supplement would support this point of view.
5. In the same spirit, why is the amplitude of the hydration layer oscillations so much smaller (1/2 o 1/3) in the presence of ions versus the pure water result in Fig 2b. Likewise after removal of the organic layer the amplitude of the hydration layer is

reduced to $\frac{1}{4}$.

6. The potential dependence of the water modes in Figure 3 is very interesting and the authors might say how reproducible the effects is when you cycle the potential over hours (as the experiments are reported to have taken place over a span of two days in the cypher ES (...after looking at the supplement this question is nicely addressed please add a comment to main text that points to this element of robustness for the sealed cell...this aspect is presently "hidden" in the supplement). How sensitive is the shift in the water modes to the identity of the supporting electrolyte. The results are shown for K₂SO₄ but what about the results for KCl or NaCl? Is the same slope obtained in the OH-1 peak position?

7. Why is the scatter band in Figure 2b for water layers KCl appear to be so much smaller than K₂SO₄. Is the difference significant, one might imagine that the halide double layer might organize in a more isotropic fashion...with regard to the water solvation and hydration layers?

8. This is very useful paper certain to help improve the robustness of the applied methods and thereby suitable for timely publication.

9. The data in the supplement clearly point to the advantages of using the closed ES cell.

10. The idea of using a nominally outer sphere redox couple to define the value of a quasi-reference electrode is wrong. The presence of the redox couple pins the Pt electrode at the E_{1/2} potential of the redox couple...which is different from the value of the Pt floating in a K₂SO₄ solution. On the scale of the potential changes examined this is a minor point...although when H₂ is generated in the cell this will tend to drive the Pt reference to H⁺/H₂ potential that is also subject to pH shear in the non-buffered electrolyte. A better way to establish the quasi-ref potential would be to compare the cyclic voltammetry in K₂SO₄ using a Ag/AgCl/Cl vs Pt and trying to minimize the extension into the HER region.

Reviewer #2

(Remarks to the Author)

This study explores the complex structure of interfacial water under realistic conditions by employing a combined approach of three-dimensional atomic force microscopy (3D-AFM) and interface-sensitive Raman spectroscopy (SHINERS). The focus is on the graphite–water interface, which is highly relevant to applications in biology, energy conversion, and environmental technologies. The authors identify three distinct interfacial water states: (1) a transient pristine water state near open circuit potential (OCP), characterized by strong hydrogen-bond (HB) breaking; (2) a hydrocarbon-dominated state, also near OCP but more stable, showing weak HB breaking due to the accumulation of hydrocarbon species; and (3) a stable pristine water state at sufficiently negative potentials, exhibiting a broader distribution of HB configurations. The complementary use of 3D-AFM for spatial liquid density profiling and SHINERS for chemical structure identification offers in situ, high-resolution insights into the interfacial water structure within ~2 nm of the surface. While the study presents a compelling framework, several important points need clarification before the manuscript can be recommended for publication:

1. The authors should present AFM images of the graphite surface and clearly indicate where the measurements were taken. Given that graphite surfaces can include step edges, terraces, and other defects—with known variations in adhesion properties—it is essential to specify whether the measurements were conducted on atomically flat terraces or at defect sites. Relevant prior studies, such as H. Lee et al., ACS Nano 9, 3814–3819 (2015) and Z. Ye et al., Appl. Phys. Lett. 106, 231603 (2015), should be cited to contextualize this variability.

2. To rule out the influence of potential adsorbates formed during electrochemical measurements, the authors should provide AFM images of the graphite surface taken both before and after the experiments. This would help confirm that the observed interfacial structures are intrinsic rather than artifacts of surface modification.

3. The mechanical pressure exerted by the AFM tip during 3D-AFM measurements is expected to be significantly higher than that in SHINERS. The authors should discuss whether this difference in applied pressure could influence the water structure near the interface and whether it could account for any observed discrepancies between the two techniques.

4. The observed shift in interlayer spacing from ~3 Å to 4–5 Å over the course of one hour warrants further explanation. First, how is the "0-minute" time point defined? Since time is required for probe alignment and approach, there may be an earlier, unobserved state preceding the nominal start of measurement. Second, considering the layered nature of graphite, tip-induced deformation or water intercalation between the top and sub-surface graphene layers may contribute to this evolution. The authors should clarify the physical origin of the time-dependent changes in interlayer spacing and discuss the potential role of tip-sample interactions or interfacial water penetration.

Reviewer #3

(Remarks to the Author)

This is an exciting fundamental study of the water/solid interface using a combination of three-dimensional atomic force microscopy and interface-sensitive Raman spectroscopy. (3D-AFM and SHINERS) It is used to probe of dynamic hydrogen bonding network at the interface to understand the role of salt and organic species within the interfacial region within 2 nm of graphite surface. They observe two interfacial configurations, transient state pristine water with strong hydrogen bonds, and another, with hydrocarbons dominating the interface, and weak hydrogen bond breaking. As far as I can tell the procedures are reasonable, though I am not an experimentalist. The insight of the observations motivates exploration of theoretical and simulation models of the phenomena observed in this complex system. Communication of the details and findings will benefit the community. Please publish.

Reviewer #4

(Remarks to the Author)

This paper studies the interfacial water structure on the HOPG using a combination of 3D-AFM and SEIRAS. While 3D-AFM has greatly advanced our understanding of the liquid-solid interface by providing visualization of the interfacial molecule arrangement, it offers limited information on the physicochemical interactions between them. This paper provides a comprehensive picture of the interfacial water structure on both pristine and hydrocarbon-adsorbed HOPG. The correlative analysis of 3D-AFM and SEIRAS bridges the gap between two contrasting models of the HOPG-water interface: majorly water at the surface and majorly adventitious non-water molecules at the surface. They demonstrate a time- and potential-dependent transition between two states. In addition, they report disrupted hydrogen bonding (HB) at the pristine HOPG-water interface, whereas the hydrocarbon-covered surface shows less broken HB. Given that the crucial roles of interfacial water in physics, biology, and especially chemistry as solvent, reactant, and proton donor to modulate the activation energy of chemical reactions, this well-written paper will be of broad interest to the readership of Nature Communication. The high-quality correlative approach could serve as a valuable framework for future studies of interfacial water structures. We therefore support publication of this manuscript after the authors address the following minor comments.

1. The authors conclude that pristine HOPG, rather than adsorbed hydrophobic molecules (probably adventitious hydrocarbons), weakened the HB network of water molecules leading lower HB-bonded or non-HB-bonded water molecules. Since HOPG itself is a carbon-based materials, what kinds of characteristic of HOPG may contribute to this difference?
2. The authors should discuss potential zero charge (PZC) of the HOPG? The direction and the strength of the interfacial electric field should be discussed with reference to the PZC (rational potential scale).
3. I recommend against defining '0-0.4 V as OCP'. This is an enormously broad potential range and should not collectively be called OCP.
4. The SHINERS particles will form their own double layer which will potentially be influenced by changes in the charge distribution imparted by the application of different potentials to the working electrode. The authors point to other references that discuss the non-interaction of these particles. However, these discussions need to be present in the current manuscript to make the data credible. Thus, relevant discussion needs to be added.
5. The authors mention that minimal changes are observed with the presence of ions but Figure 2b shows considerable changes in the force curves. These changes need to be more clearly discussed and elaborated.
6. How do the authors distinguish hydrocarbon adsorption from potential slow formation of oxygen groups on the HOPG surface that could also lead to structural changes?

It would have been nice to show measurements where organics are purposefully added to the electrolyte to corroborate the claim that the changes in force curves occur because of organics adsorption. I realize, however, that this may be beyond the scope of the study.

Reviewer #5

(Remarks to the Author)

Version 1:

Reviewer comments:

Reviewer #1

(Remarks to the Author)

The authors have done a nice job addressing my questions. The paper is suitable for publication.

Reviewer #2

(Remarks to the Author)

I have carefully reviewed the revised manuscript and the authors' response letter. The authors have addressed my comments and concerns appropriately, and the manuscript has been significantly improved. I therefore recommend that the paper be accepted in its current form.

Reviewer #5

(Remarks to the Author)

Title: Correlative angstrom-scale microscopy and spectroscopy of graphite–water interfaces

Manuscript ID: NCOMMS-25-49442-T

Authors: Lalith Krishna Samanth Bonagiri, Diana M. Arvelo, Fujia Zhao, Jaehyeon Kim, Qian Ai, Shan Zhou, Kaustubh S. Panse, Ricardo Garcia, and Yingjie Zhang.

Reply to the Editor & All Reviewers

We would like to thank the editor for handling our manuscript and for the valuable suggestions on format changes. We have now thoroughly addressed all the reviewers' comments, revised/added some texts in the main manuscript, and added Supplementary Note 1 and Supplementary Figs. 1, 2, 15, 16, 28, 30b, and 31 to the Supplementary Information. In addition, we have added “Data availability”, “Acknowledgements”, “Author contributions”, and “Competing interests” sections at the end of the main manuscript. To facilitate the broad dissemination of our results, we have attached an Excel file containing all the source data presented in the main manuscript and supplementary information, and another Excel file containing all the raw Raman spectral data (directly produced by the Raman instrument, before any data processing step). We have also attached the required reporting summary form. Point-by-point responses to the reviewers' comments are provided below. We believe this manuscript is now at a publication-ready level.

Reply to Reviewer 1

Reviewer comment 1.0: *This is an excellent paper that combines AFM and SHINERS to investigate the hydration layers adjacent to HOPG surfaces in a variety of different, but nominally neutral pH, electrolytes. The core of the paper is aimed at resolving ambiguities associated with the observed spacing between hydration layers that have been observed to vary between ~ 0.3 nm and ~ 0.5 nm in different reports in the literature that include different or unspecified preparation conditions. By a combination of aging studies under open circuit conditions and polarization studies at negative potential the authors attribute the observation of thicker hydration layer to organic contamination that tends to build up with time under open circuit conditions. The authors show that this layer can be removed at negative potentials which is an interesting and helpful finding. Likewise, an important aspect of the work is that the contributions follow from interactions and parallel experiments performed between two independent laboratories that represent leading practitioners in fast force methods for probing the hydration layers at immersed interfaces. The effectiveness of sealing the cell is suggested by data in the supplement and perhaps more should be made of this in the main text. The authors provide Tables listing the imaging parameters in the supplement that will definitely help others interested in reproducing this work.*

The paper is certainly worthy of publication. That said, the authors may wish to consider the following comments prior to going forward with publication.

Response 1.0: We would like to thank the reviewer for the thoughtful and constructive comments. We highly appreciate the recognition of the significance of our work and careful reading of the manuscript and supplementary information. The reviewer's insights have been very helpful in clarifying and strengthening the presentation of our results. We have carefully considered the

suggestions and have made the necessary revisions to the manuscript accordingly. Please find our detailed point-by-point responses below.

Reviewer comment 1.1: *In the case of the SHINERS experiments the preparation routine involved allowing the SHINERS particles to dry in the ambient for 1 to 2 hours. Did the surface become covered with organic contamination under these conditions? If not, does this say something about the difference between gas phase versus condensed phase exposure? Please clarify the history of the SHINERS experiments in this regard.*

Response 1.1: The question involves both the SHINERS sample preparation routine and broader contamination mechanisms. Our answers are:

- 1) Surface condition after ambient drying in SHINERS experiments.** In SHINERS measurements, including both our work and reports in the existing literature (e.g., *ACS Nano* 2013, 7, 8940–8952), the surface commonly becomes covered with organic contaminants after the drying step. This contamination is immediately evident in the Raman spectra, which display characteristic C–H stretching bands (2800–3000 cm^{-1}), as well as C–C and C=C modes typical of adventitious carbon (Fig. 4c and Supplementary Figs. 25a,b and 26a,b.). It is challenging to control the contamination mechanism in this drying step, as this process does not correspond to either pure gas-phase or pure liquid-phase exposure; during evaporation, a thin water layer persists on the surface, creating a semi-wet state. Instead of trying to control the contamination source during drying, our strategy was to “initialize” the sample condition after the SHINERS sample has been prepared. As discussed in Page 8 | Lines 7–10 of the original manuscript (Page 8 | Lines 23–26 after revision), we applied reductive (negative) polarization to clean the surface, as the adsorbed organic species were removed under these conditions. After this initiation procedure, we restored a pristine graphite–water interface. All the following air exposure and electrode potential application history were well documented in our data and reported in the manuscript (Figs. 2c, 3c, and 4c, Supplementary Figs. 23–27), ensuring the results correspond to known experimental conditions and can be reproduced by others.
- 2) Gas-phase vs condensed-phase exposure: context from prior work.** This question relates to the broader issue of contamination pathways under gas- versus liquid-phase exposure. Haitao Liu’s group has examined these differences extensively, combining electrochemical measurements with ellipsometry and contact angle analysis (*Carbon* 2018, 134, 464–469; *Nat. Mater.* 2013, 12, 925–931). Their studies demonstrated that both direct ambient air exposure and air exposure of the solid immersed in aqueous solution/water (in the time scale of minutes) are sufficient to generate measurable contamination layers on the surface. The determination of the exact kinetics of the gas-phase vs liquid-phase air exposure is beyond the scope of this work.

Changes to the manuscript: In the “Methods” section of the main manuscript, we further clarified the electrochemical cleaning procedures for SHINERS sample preparation (Page 18 | Lines 9–10): “During drying, the surface usually accumulated organic contaminants, which were subsequently removed by negative electrode polarization (electrochemical cleaning).”

Reviewer comment 1.2: *What is the prospect of intercalation reactions, e.g. proton or one of the other cations, during the excursion to negative potentials? Does the hysteresis between the water spectra e.g. the OH-1 topwater layers, Figure 2c, Figure 3c at 0 V versus that broadened and shifting OH-1 following the step to negative potentials in Figure 4C and returning to 0 V, speak to an irreversible component in the cleaning process. Could this be related to intercalation. The authors might wish to consider XPS analysis of the HOPG to see if the cation insertion is doing something to the HOPG under the conditions considered.After reading the supplement I see Figure 19 nicely addresses the hysteresis question and perhaps this should be mentioned in the main text.*

Response 1.2: In graphite, the potentials required for intercalation of protons or other cations (e.g., Li^+ / Na^+ / K^+ at -2.5 V to -3 V vs. SHE) lie beyond the stability window of water (see results in *Sci. Rep.* 2016, 6, 28421; *Adv. Sci.* 2017, 4, 1700146). As a result, in aqueous electrolytes cation intercalation into HOPG is generally not observed. Consistent with existing studies, in our cyclic voltammetry measurements we do not detect any peaks—either reversible or irreversible—that would be indicative of cation or proton intercalation (see Supplementary Fig. 22).

As the reviewer mentioned, Supplementary Fig. 19 in the original manuscript (Supplementary Fig. 23 after revision) proves that the hysteresis in the 0 V spectra is not due to any other reason except the hydrocarbon removal/accumulation effects. To emphasize this point, we have added an additional comment in the main manuscript referring to this figure.

Changes to the manuscript: In the “**Electrified graphite–water interfaces: non-pristine response**” section of the main manuscript, we added (Page 14 | Lines 23–26): “*Additionally, the change in the initial and final 0 V spectra in Fig. 4c cannot be explained by other potential-induced hysteresis effects besides the change of hydrocarbons, as the pristine, water-dominated interface did not exhibit any hysteresis effect (Supplementary Fig. 23).*”

Reviewer comment 1.3: *Did you track the current transients associated with the potential step experiments? Is the conversion from the organic contaminated state charge dependent? Are the gas bubbles necessary to remove the organic contaminants whereby the gas/electrolyte interface surface area serves to scavenge that at the HOPG interface. Do the current transient for the potential steps reveals the presence of the contamination layers? After looking though the supplement I see the voltammograms...but the variation stated in the text are not evident. So the question remains...as to whether or not chronoamperometry of voltammetry can be used to probe for a clean surface.*

Response 1.3: We agree that this is an important point, and we did in fact track the current transients associated with the potential step experiments in both Urbana and Madrid laboratories. In both labs, potential steps performed on hydrocarbon-contaminated HOPG/solution interface produced markedly smaller current responses than those on a pristine interface (see the added Supplementary Fig. 28). The pristine interface consistently exhibited larger current transients than those of the contaminated one, revealing that the former has higher double layer capacitance than the latter. This is likely because the hydrocarbons accumulated at the HOPG–solution interface induce a larger spatial separation between the electrical double layers and the HOPG surface, which further leads to smaller capacitance. Overall, these observations suggest that the transient current may indeed serve as a qualitative probe of surface cleanliness. However, more systematic

measurements will be required to establish the current–cleanliness relationship quantitatively, which is beyond the scope of this work.

Regarding removal of hydrocarbons, we note that our AFM and SHINERS measurements were limited to potentials below the onset of visible bubble formation, since strong gas evolution would disrupt stable 3D-AFM imaging and SHINERS spectroscopy. This was discussed in the original manuscript (originally Page 7 | Lines 42–45, now Page 8 | Lines 12–15) “*At sufficiently negative potentials, hydrogen evolution reaction (HER) became significant (Supplementary Fig. 22), producing bubbles in the local area that prevented further 3D-AFM and SHINERS measurements.*” Since hydrocarbon removal was observed using 3D-AFM and SHINERS measurements, we can rule out the possibility that large, visible bubble formation led to hydrocarbon removal. However, we cannot rule out the possibility that small bubbles or hydrogen molecules (formed via hydrogen evolution reaction) can be (partially) responsible for hydrocarbon removal. We have discussed a few possible origins for the potential-induced hydrocarbon removal mechanisms in the paragraph right before “**Conclusion and outlook**” in the original manuscript, which we believe is sufficient for the scope of this work.

Changes to the manuscript:

1) We added a paragraph in the main manuscript to explain the current transient measurements (Page 14 | Lines 28–31): “*As another indicator of the interfacial liquid structure, we recorded the current transients during SHINERS measurements (Supplementary Fig. 28). The water-dominated interface exhibited larger transient currents than those of the hydrocarbon-dominated one, due to the expected higher double layer capacitance of the pristine interface.*”

2) We added a figure in the Supplementary Information, as reproduced below:

Supplementary Fig. 28 | Current transients recorded during SHINERS measurements (August 10, 2021, Urbana). Gas environment: air; open. Potential reference: Ag/AgCl. The pristine and non-pristine data were recorded during SHINERS measurements shown in Supplementary Fig. 23 (potential cycle 3) and Supplementary Fig. 26 (potential cycle 1), respectively. The electrode potential was changed at a step of -0.2 V , and was held constant (at the values specified on each panel) while the current transient was recorded. The pristine interface consistently exhibited larger current transients than those of the non-pristine one, revealing that the former has higher double layer capacitance than the latter. This is likely because the hydrocarbons accumulated at the HOPG–solution interface induce a larger spatial separation between the electrical double layers and the HOPG surface, which further leads to smaller capacitance. Overall, these observations suggest that the transient current may serve as a qualitative indicator of surface cleanliness.

Reviewer comment 1.4: *In the introduction the statement is made that the hydration layer are independent of the identity of the supporting electrolyte even at concentrations in the range of 0.1 to 1 mol/L. This seems very unintuitive to me. In looking at the variation in d_{12} across the figure set, e.g. 0.29, 0.28, 0.33 nm in Figure 2, perhaps the claim of independence of d_{12} from ion identity is too strong a statement and rather the differences are yet to be revealed or understood due to sampling, analysis, entropic effects. My sense is the variation evident in the supplement would support this point of view.*

Response 1.4: We fully agree that the error bars of experimental results need to be thoroughly analyzed, before any conclusion on the electrolyte-dependence can be reached. Thanks to the significant amounts of data we have generated, we have been able to thoroughly quantify the average values and error bars of d_{12} with different ion types and concentrations. As shown in the added Supplementary Fig. 15, we can conclude that d_{12} is insensitive to the supporting electrolyte within the error range of our experimental results. Of course, below the experimental error range (roughly 0.2 Å), we cannot make any statement about the electrolyte-dependence.

Changes to the manuscript:

1) The original statement in the main manuscript Page 4 | Lines 22–23, “*This interpretation is consistent with prior reports that showed the salt-independence of hydration layer spacings in dilute solutions (<1 M)*”, is now changed to (Page 4 | Lines 28–31) “*This interpretation is consistent with prior reports suggesting that hydration layer spacings are largely insensitive to the identity and concentration of the supporting electrolyte in dilute solutions (<1 M), since the number density of water is much higher than that of the ionic species in these solutions*”.

2) We added a new paragraph in the main manuscript to further clarify the error analysis and potential origins of experimental errors (Page 4 | Lines 39–46): “*Additionally, care must be taken to avoid overinterpreting differences in the observed 3D-AFM force curves. Random fluctuations, due to variations in AFM tip condition, environmental factors and instrument noise (considering the two different locations and four-year time span), can result in various error ranges in the experimental observables. For example, average d_{12} values within 2.8–3.3 Å were observed for pristine HOPG–aqueous solution interfaces with different ionic species and concentrations (Supplementary Fig. 15). Force oscillation amplitudes, on the other hand, exhibits larger fluctuations between 0.01–0.1 nN (Supplementary Fig. 16). These variations should not be directly attributed to differences in the electrolytes or ageing/cleanliness conditions.*”

3) We added Supplementary Fig. 15:

Supplementary Fig. 15 | Summary of interlayer distances (d_{12}) obtained from multiple 3D-AFM measurements at the pristine HOPG–aqueous solution interface across different electrolyte compositions (ion types and concentrations) and electrode potentials. For pure water, the 0-min data from all measured datasets (M1, M2, and M3) were used (corresponding to Supplementary Fig. 3a–c). For 0.1 M KCl, M4 (0 min, post-scan aging at 0, 30, 60, 90, 120 min), M5, and M10 (–1.78 V) were used (Supplementary Figs. 3d, 14a, 11a, and 13c). For 0.1 M

K_2SO_4 , datasets U4 and U6 were used (Supplementary Figs. 8 and 10). For 0.2 M K_2SO_4 , datasets M8 (−1.28 V, −1.78 V, and post-scan aging at 20, 40, 60 min) and M9 (−1.28 V, −1.78 V) were used (Supplementary Figs. 13a,b and 14b). For 0.01 M K_2SO_4 , datasets U1, U2 (including potential scan 1: −1.6 V, −2.2 V, −2.6 V), U3 (including potential scan 1: −1.8 V), U5, M6, and M7 were used (Supplementary Figs. 5–7, 9, 11b,c, and 12a,b).

Reviewer comment 1.5: *In the same spirit, why is the amplitude of the hydration layer oscillations so much smaller (1/2 or 1/3) in the presence of ions versus the pure water result in Fig 2b. Likewise after removal of the organic layer the amplitude of the hydration layer is reduced to 1/4.*

Response 1.5: Similar to the error analysis of d_{12} as stated in Response 1.4, we conducted additional thorough error analysis of the force oscillation amplitude, in order to verify whether the variations of this metric are due to the inherent sample structure or external, uncontrollable factors. These additional analyses are shown in the added Supplementary Fig. 16. The conclusion is that these variations are due to external factors (e.g., random variations in AFM tip geometry for different measurements).

Changes to the manuscript:

- 1) Same changes in Page 4 | Lines 28–31 and Page 4 | Lines 39–46 as stated in Response 1.4.
- 2) We added Supplementary Fig. 16:

Supplementary Fig. 16 | Force oscillation amplitude analysis of 3D-AFM datasets. *a*, A representative 3D-AFM dataset (corresponding to Supplementary Fig. 10a, −0.8 V) together with schematics depicting the extraction of the force oscillation amplitude. *b,c*, Summary of measured force oscillation amplitudes from representative 3D-AFM datasets as a function of electrode potential for (b) pristine HOPG–electrolyte interfaces (corresponding to Supplementary Figs. 5–10, 11a, and 11c) and (c) non-pristine interfaces (corresponding to Supplementary Fig. 13) with different electrolyte compositions and concentrations. The overall spread in force values exceeds any systematic dependence on electrolyte concentration or ion identity.

Reviewer comment 1.6: *The potential dependence of the water modes in Figure 3 is very interesting and the authors might to say how reproducible the effects is when you cycle the potential*

over hours (as the experiments are reported to have taken place over a span of two days in the cypher ES (...after looking at the supplement this question is nicely addressed please add a comment to main text that points to this element of robustness for the sealed cell...this aspect is presently “hidden” in the supplement). How sensitive is the shift in the water modes to the identity of the supporting electrolyte. The results are shown for K₂SO₄ but what about the results for KCl or NaCl? Is the same slope obtained in the OH-1 peak position?

Response 1.6: We agree that argon sealing is effective in maintaining a more stable interface with slower ageing. However, this does not mean that experiments carried out in air are not reproducible or do not enable the observation of the intrinsic potential-dependence. In the original manuscript, at Page 8 | Lines 17–20 (Page 8 | Lines 33–36 in the revised manuscript), we mentioned that “*To minimize the possible convolution of ageing to the potential dependence of the interfacial water structure, the measurements were conducted either quickly (within ~20 minutes for each potential scan) to mitigate ageing or in an argon-sealed liquid cell to slow down the ageing process.*” We believe this is an accurate statement regarding the effect of argon sealing. To further clarify this effect, we added another sentence right after the above description in the revised manuscript): “*As shown in Supplementary Figs. 5–10, argon sealing was usually effective in preserving the pristine interface for at least 30 hours.*”

Regarding the dependence of the OH stretching peak position on electrolyte identity, we agree that this is an important question. Previous studies proposed some weak electrolyte-dependence of the interfacial water vibrational modes (*Nat. Mater.* 2019, 18, 697–701; *Nature* 2021, 600, 81–85; *Joule* 2023, 7, 1652–1662), although a reliable conclusion cannot be reached due to the lack of comprehensive control experiments and quantitative analyses. In the past four years, we have attempted a few SHINERS experiments on different electrolytes. However, similar to the 3D-AFM results (Supplementary Fig. 15), we are not able to reach a comprehensive conclusion yet about the electrolyte-dependence, due to the random fluctuations induced by external experimental factors. We will continue such efforts in the future, but we believe a comprehensive study of electrolyte-dependence is beyond the scope of this paper, which is mainly focused on the impact of environmental ageing and electrode potential on the interfacial water structure.

Changes to the manuscript: at Page 8 | Lines 36–37, we added: “*As shown in Supplementary Figs. 5–10, argon sealing was usually effective in preserving the pristine interface for at least 30 hours.*”

Reviewer comment 1.7: *Why is the scatter band in Figure 2b for water layers KCl appear to be so much smaller than K₂SO₄. Is the difference significant, one might imagine that the halide double layer might organize in a more isotropic fashion...with regard to the water solvation and hydration layers?*

Response 1.7: As explained in Responses 1.4 and 1.5, we should avoid overinterpreting differences in the observed 3D-AFM force curves, due to random fluctuations induced by AFM tip condition, environmental factors and instrument noise (considering the two different locations and four-year time span). We believe Responses 1.4 and 1.5 and the manuscript revisions specified therein already addressed this concern. To further examine the direct comparison between KCl and K₂SO₄ solutions, we have prepared Fig. R1 below using the existing data reported in the

manuscript. We can see that the difference in electrolyte is not the reason why the scatter bands are different in Fig. 2b.

Fig. R1. Representative 3D-AFM force–distance curves showing the variation in scatter bands obtained from repeated measurements on pristine HOPG–electrolyte interfaces.

Reviewer comment 1.8: *This is very useful paper certain to help improve the robustness of the applied methods and thereby suitable for timely publication.*

Response 1.8: We concur.

Reviewer comment 1.9: *The data in the supplement clearly point to the advantages of using the closed ES cell.*

Response 1.9: We agree. Related changes in the manuscript have been specified in Response 1.6.

Reviewer comment 1.10: *The idea of using a nominally outer sphere redox couple to define the value of a quasi-reference electrode is wrong. The presence of the redox couple pins the Pt electrode at the E1/2 potential of the redox couple...which is different from the value of the Pt floating in a K2SO4 solution. On the scale of the potential changes examined this is a minor point. Although when H2 is generated in the cell this will tend to drive the Pt reference to H+/H2 potential that is also subject to pH shear in the non-buffered electrolyte. A better way to establish the quasi-ref potential would be to compare the cyclic voltammetry in K2SO4 using a Ag/AgCl/Cl vs Pt and trying to minimize the extension into the HER region.*

Response 1.10: We appreciate the reviewer’s insightful suggestions. We agree that using an outer-sphere redox couple to define the potential of a Pt quasi-reference electrode is not the most precise. As the reviewer also noted, this is a minor point considering the large range of the potential changes examined here. We also considered the suggested approach of using cyclic voltammetry in K_2SO_4 solution to establish the Pt potential. However, this method is not applicable in our system because:

- The CV of the graphite/aqueous interface is essentially featureless until the onset of hydrogen evolution reaction (HER) (see Supplementary Fig. 22).
- The HER onset potential is highly variable depending on the exact surface condition and therefore unreliable.
- HER can additionally pin the Pt reference to H^+/H_2 , which is further shifted by local pH gradients in the non-buffered electrolyte.

To further examine the possible errors in the Pt potential calibration, we performed additional control experiments in which the open-circuit potential of Pt was monitored against a stable Ag/AgCl reference electrode in 0.1 M K_2SO_4 (Supplementary Fig. 30b). The obtained OCP offset is 0.27 V, which is close to the 0.22 V obtained using the redox couple calibration. We therefore emphasize that while small variations of $\sim\pm 0.1$ V can be expected, they remain minor and do not affect the conclusions of this work.

Changes to the manuscript:

1) In Page 18 | Lines 29–31 of the main manuscript, we added “*It is worth noting that the potential of a Pt quasi-reference electrode may not be exactly fixed; variations in the rough range of ± 0.1 V may arise but do not affect the conclusions we reached in this work.*”

2) We added Supplementary Fig. 30b:

Supplementary Fig. 30 | Reference electrode calibration. a, Cyclic voltammetry curves of glassy carbon electrode / 0.1 M K_2SO_4 + 50 mM $K_4[Fe(CN)_6]$ + 50 mM $K_3[Fe(CN)_6]$ in water. Counter electrode: platinum. Reference electrode (RE): Ag/AgCl (shown in red) or platinum (shown in blue). Working electrode surface area: 19.64 mm². The potential of Pt was found to be 0.22 V more positive than Ag/AgCl. **b,** OCP recorded as a function of time for a platinum wire electrode versus an Ag/AgCl reference electrode in 0.1 M K_2SO_4 aqueous solution. The average OCP value was found to be ~ 0.27 V.

Reply to Reviewer 2

Reviewer comment 2.0: *This study explores the complex structure of interfacial water under realistic conditions by employing a combined approach of three-dimensional atomic force microscopy (3D-AFM) and interface-sensitive Raman spectroscopy (SHINERS). The focus is on the graphite–water interface, which is highly relevant to applications in biology, energy conversion, and environmental technologies. The authors identify three distinct interfacial water states: (1) a transient pristine water state near open circuit potential(OCP), characterized by strong hydrogen-bond (HB) breaking; (2) a hydrocarbon-dominated state, also near OCP but more stable, showing weak HB breaking due to the accumulation of hydrocarbon species; and (3) a stable pristine water state at sufficiently negative potentials, exhibiting a broader distribution of HB configurations. The complementary use of 3D-AFM for spatial liquid density profiling and SHINERS for chemical structure identification offers in situ, high-resolution insights into the interfacial water structure within ~2 nm of the surface. While the study presents a compelling framework, several important points need clarification before the manuscript can be recommended for publication:*

Response 2.0: We thank the reviewer for the careful evaluation and constructive comments. The feedback has been highly valuable for further strengthening the manuscript. Point-by-point responses are provided below.

Reviewer comment 2.1: *The authors should present AFM images of the graphite surface and clearly indicate where the measurements were taken. Given that graphite surfaces can include step edges, terraces, and other defects—with known variations in adhesion properties—it is essential to specify whether the measurements were conducted on atomically flat terraces or at defect sites. Relevant prior studies, such as H. Lee et al., ACS Nano 9, 3814–3819 (2015) and Z. Ye et al., Appl. Phys. Lett. 106, 231603(2015), should be cited to contextualize this variability.*

Response 2.1: We agree that the surface condition is an important factor to consider. All AFM studies presented in this manuscript were performed on atomically flat terrace sites. To further clarify this, we have added an atomic-resolution image showing the graphitic lattice (Supplementary Fig. 1), where 3D-AFM measurements were subsequently carried out. The behavior of interfacial structures at edge and other defect sites is beyond the scope of the present study. However, in a separate study, we have found exciting emergent effects of a series of solid–liquid interfaces at defect sites, which we will submit as another manuscript. We have added statements in the main manuscript to clarify the measurement sites and cited the references (Refs. 38 and 39) suggested by the reviewer.

Changes to the manuscript:

1) On Page 3 | Lines 30–31 & Page 4 | Lines 1–2 of the main manuscript, we have revised the original simplified statements on 3D-AFM imaging on HOPG surface to: *“All 3D-AFM measurements were performed on atomically flat terrace sites of HOPG immersed in water or aqueous solution (Supplementary Fig. 1). The HOPG surface remained stable over time and after potential cycles (Supplementary Fig. 2). While step edges exist on the HOPG surface^{38,39}, those sites were not examined by 3D-AFM in this work.”*

2) We added Supplementary Fig. 1:

Supplementary Fig. 1 | AFM phase image of a pristine HOPG surface immersed in water. All 3D-AFM measurements in this work were conducted in regions similar to the one depicted here. The lattice-resolution image reveals a hexagonal lattice with a lattice constant of ~ 2.5 Å. The image was obtained hundreds of nm away from any step edge. Imaging was performed at Madrid, using PPP-FM cantilevers (Nanosensors) operated at the second eigenmode (AC, amplitude modulation mode). Experimental parameters: free amplitude, 200 pm; setpoint amplitude, 150 pm; resonant frequency, 195.652 kHz; spring constant, 100 N m^{-1} ; quality factor, 10.5.

Reviewer comment 2.2: To rule out the influence of potential adsorbates formed during electrochemical measurements, the authors should provide AFM images of the graphite surface taken both before and after the experiments. This would help confirm that the observed interfacial structures are intrinsic rather than artifacts of surface modification.

Response 2.2: We have now performed the experiment suggested by the reviewer. We imaged a graphite surface immersed in a 200 mM K_2SO_4 solution before and after the application of -1 V (vs Pt). A comparison of tapping mode AFM images before and after the application of this potential did not show any obvious difference (Supplementary Fig. S2).

Changes to the manuscript:

1) At Page 3 | Line 31 & Page 4 | Line 1 of the main manuscript, we added “The HOPG surface remained stable over time and after potential cycles (Supplementary Fig. 2).”

2) We added Supplementary Fig. 2:

Supplementary Fig. 2 | Large-area AFM images of HOPG surface in aqueous solution. Location: Madrid. Electrolyte: 200 mM K_2SO_4 in water. HOPG surface topography maps were measured at 0 V, (a) before and (b) after applying -1 V vs Pt for 2 min. The cross-sectional height profiles are shown below the corresponding images, indicating no change of the step height. Imaging mode: AC, amplitude modulation, first eigenmode. AFM cantilever: ArrowUHFuA (Nanosensors). Imaging

parameters: resonant frequency, 423.94 kHz; spring constant, 5.35 N m^{-1} ; quality factor, 4.2; free amplitude, 3.8 nm; setpoint amplitude, 2.73 nm.

Reviewer comment 2.3: *The mechanical pressure exerted by the AFM tip during 3D-AFM measurements is expected to be significantly higher than that in SHINERS. The authors should discuss whether this difference in applied pressure could influence the water structure near the interface and whether it could account for any observed discrepancies between the two techniques.*

Response 2.3: We appreciate the reviewer’s insightful comment regarding the possible effect of tip pressure in 3D-AFM compared to SHINERS. We address this in the following sections, considering (i) the possible deformation of the graphite substrate under the applied load, and (ii) the possible influence of the AFM tip on the interfacial liquid structure.

1. Graphite deformation under applied load:

To evaluate whether the mechanical load in our measurements could perturb the interfacial water structure, we estimated the elastic deformation of a graphite surface under an AFM tip using the Hertz contact model. The effective modulus E^* of a silicon probe interacting with a graphite substrate is given by

$$\frac{1}{E^*} = \frac{1 - \nu_t^2}{E_t} + \frac{1 - \nu_s^2}{E_s}$$

where Young’s modulus $E_t \approx 160$ GPa and Poisson ratio $\nu_t \approx 0.278$ for silicon (*J. Microelectromech. Syst.* 2010, 19, 229–238; *Nat. Commun.* 2017, 8, 1944) and $E_s \approx 36$ GPa and $\nu_s \approx 0.24$ for graphite (*Nat. Commun.* 2017, 8, 1944; *Phys. Rev. B* 2007, 75, 153408), yielding $E^* \approx 31$ GPa. Under typical 3D-AFM measurement conditions (tip radius $R \approx 5$ nm, maximum load of $F \approx 500$ pN), the maximum indentation depth is $\delta = \left(\frac{3F}{4E^*\sqrt{R}}\right)^{2/3} \approx 0.31$ Å, which is substantially smaller than the thickness of a single interfacial water layer (≈ 3 Å). Therefore, the effect of the mechanical deformation in the 3D-AFM data is negligible.

Furthermore, the pressure-induced substrate deformation, if any, is only expected to occur after the probe penetrates through the first interfacial water layer. Thus, any elastic deformation would only influence the spacing between the observed substrate surface and the first liquid layer, and have no impact on the separation between the upper liquid layers. Since our results concern the periodic structuring of the liquid layers, all our analyses will not change even if the substrate do have some deformation when the tip reaches the very end of the approach curve.

In a separate study, we have conducted 3D-AFM imaging of an aqueous solution at an HOPG step edge site. As shown in Fig. R2 below, we observed an HOPG step height of ~ 3 Å and hydration layer spacing of ~ 3 Å, both consistent with standard or expected intrinsic structure of HOPG and water. We thus conclude that the small force used in 3D-AFM does not result in observable structural distortions of the whole interface.

[redacted]

2. Effect of the tip on the interfacial liquid structure:

It is important to clarify that the oscillatory forces measured in 3D-AFM do not arise from mechanical compression of water molecules. Instead, those forces reflect the local configurational free energy of interfacial water, which is largely entropic in nature. Our recent perspective (*J. Phys. Chem. C* 2025, 129, 5273–5286) discussed this mechanism extensively. The observed force oscillations represent probe-induced modulations of interfacial liquid configurational entropy, rather than a hydrostatic pressure. Unlike in surface force apparatus (SFA) measurements where two extended surfaces strongly confine the liquid, 3D-AFM probes a localized volume in which water molecules remain free to diffuse and reorganize dynamically. Consequently, the oscillatory forces correspond directly to the intrinsic liquid density oscillation at the interface.

In addition, experimental evidence supports this interpretation: in Figure 9 of *J. Phys. Chem. C* 2025, 129, 5273–5286 (reproduced below), nearly identical oscillatory force curves were obtained using AFM tips with radii of 10 nm and 250 nm, demonstrating that the observed solvation forces do not depend on mechanical indentation. Similarly, our recent work (*Proc. Natl. Acad. Sci. U.S.A.* 2025, 122, e2421635122; Fig. S5) showed analytically that only the terminal water molecule(s) adjacent to the tip contributes to the measured solvation force, which can be expressed as a function of the intrinsic, unperturbed local liquid density. Thus, 3D-AFM captures the entropic reorganization of interfacial water rather than its mechanical compression.

J. Phys. Chem. C 2025, 129, 5273–5286; **Fig. 9**

Proc. Natl. Acad. Sci. U.S.A. 2025, 122, e2421635122; **Fig. S5**

Changes to the manuscript:

1) At Page 2 | Lines 36–37 of the main manuscript, we added: “On the other hand, 3D-AFM captures the local liquid density^{30,33} without inducing observable substrate deformations (Supplementary Note 1).”

2) We added Supplementary Note 1:

Using the Hertz contact mechanics model, the deformation (indentation) of the graphite substrate under our imaging conditions can be estimated. The effective elastic modulus E^* for a silicon tip in contact with graphite is

$$\frac{1}{E^*} = \frac{1 - \nu_t^2}{E_t} + \frac{1 - \nu_s^2}{E_s},$$

where Young's modulus $E_t \approx 160$ GPa and Poisson ratio $\nu_t \approx 0.278$ for silicon^{1,2}. For graphite, these values are $E_s \approx 36$ GPa and $\nu_s \approx 0.24$ ^{2,3}. Inserting these values into the above formula, we obtain $E^* \approx 31$ GPa. For a spherical tip of radius R under a normal load F , the indentation δ is given by the Hertz relation:

$$\delta = \left(\frac{3F}{4E^*\sqrt{R}} \right)^{2/3}$$

Using representative 3D-AFM parameters (tip radius $R \approx 5$ nm, maximum load of $F \approx 500$ pN), the indentation depth is $\delta \approx 0.31$ Å. This value is substantially smaller than the interlayer distance of pristine interfacial water (~ 3 Å).

Reviewer comment 2.4: *The observed shift in interlayer spacing from ~ 3 Å to 4–5 Å over the course of one hour warrants further explanation. First, how is the “0-minute” time point defined? Since time is required for probe alignment and approach, there may be an earlier, unobserved state preceding the nominal start of measurement. Second, considering the layered nature of graphite, tip-induced deformation or water intercalation between the top and sub-surface graphene layers may contribute to this evolution. The authors should clarify the physical origin of the time-dependent changes in interlayer spacing and discuss the potential role of tip-sample interactions or interfacial water penetration.*

Response 2.4: We appreciate the reviewer's quest for a thorough evaluation of alternative explanations for the evolution of liquid layer spacing. These points are addressed below.

1. Definition of time-zero and initial interfacial structure:

Importantly, the ~ 3 Å interlayer spacing corresponds to well-established intrinsic layer spacing in pure water (see previous reports: *Nat. Commun.* 2016, 7, 12164; *Nat. Commun.* 2017, 8, 2111; *ACS Nano* 2012, 6, 9013–9020; *Proc. Natl. Acad. Sci. U.S.A.* 2024, 121, e2407877121). Thus, defining this point as “time zero” is physically justified—the system is at its intrinsic baseline condition, and the precise elapsed time since sample preparation is not critical to the interpretation of the subsequent evolution. We are not aware of any theoretical predictions of an interlayer spacing smaller than 3 Å for interfacial water, and have never observed such small spacings (beyond the experimental error range) at HOPG–water interface regardless of how fast we prepare the sample and probe. Therefore, it is safe to assume that there is no “unobserved state preceding the nominal start of measurement.”

In the actual experiments, the total duration from graphite exfoliation (to expose a pristine surface) to probe alignment and initiation of 3D-AFM measurement is in the scale of 5 minutes, which is almost negligible compared to the 1–2 hours required to observe the 3 Å to 4–5 Å shift in liquid layer spacing. For this reason, we specified in the original manuscript that

(originally Page 8 | Lines 8–10, now Page 8 | Lines 24–26): “*State 1 was achieved by either limiting the total sample preparation/air exposure time to no more than a few minutes before starting the measurement, or via electrochemical cleaning after the sample was prepared (as discussed later).*” The “0-minute” time point corresponds to the time when the first 3D-AFM force map was acquired for the pristine HOPG–water interface.

2. Tip-induced deformation and intercalation:

We have thoroughly discussed possible tip-induced deformations in Response 2.3. Briefly, our calculations indicate that graphite deforms less than 0.3 Å throughout the whole 3D-AFM measurement process, which has negligible impacts on the conclusions. Our measurements at HOPG step edges revealed a constant step-height at ~3 Å (Fig. R2), eliminating any possibility of intercalation effects. Furthermore, electrochemical intercalation of ions or protons is not feasible under our conditions, as the required insertion potentials (~ –2.5 to –3 V vs SHE) lie beyond the range of our measurements (see results in *Sci. Rep.* 2016, 6, 28421; *Adv. Sci.* 2017, 4, 1700146). Consistent with existing studies, in our cyclic voltammetry measurements we did not detect any peaks that would be indicative of any intercalation effects (see Supplementary Fig. 22).

As to the reviewer’s request that “*The authors should clarify the physical origin of the time-dependent changes in interlayer spacing*”, this was already clarified in Page 4 | Lines 2–5 of the original manuscript (Page 4 | Lines 8–10 in the revised manuscript): “*The observed time-evolution is consistent with previous reports^{6,25,40}. As explained before, ~3 Å corresponds to pristine hydration layers, while 4–5 Å is likely due to the accumulation of adventitious species whose exact nature remains unknown to date^{6,25,40}.*” This explanation is further corroborated by our SHINERS measurements revealing the accumulation of hydrocarbons after ageing in air (Fig. 2c) and our control experiments where the ~3 Å spacing is preserved over 1–2 days in an electrochemical cell sealed in argon gas (Supplementary Figs. 5–10).

Changes to the manuscript:

1) The same changes specified in Response 2.3 that addressed the concern on possible substrate deformation.

2) At Page 4 | Lines 6–8 of the main manuscript, we added: “*The “0 min” time point corresponds to the time when the first 3D-AFM data frame was acquired at the pristine HOPG–water interface.*”

Reply to Reviewer 3

Reviewer comment 3.0: *This is an exciting fundamental study of the water/solid interface using a combination of three-dimensional atomic force microscopy and interface-sensitive Raman spectroscopy. (3D-AFM and SHINERS) It is used to probe of dynamic hydrogen bonding network at the interface to understand the role of salt and organic species within the interfacial region within 2 nm of graphite surface. They observe two interfacial configurations, transient state pristine water with strong hydrogen bonds, and another, with hydrocarbons dominating the interface, and weak hydrogen bond breaking. As far as I can tell the procedures are reasonable,*

though I am not an experimentalist. The insight of the observations motivates exploration of theoretical and simulation models of the phenomena observed in this complex system. Communication of the details and findings will benefit the community. Please publish.

Response 3.0: We thank the reviewer for the evaluation of our work from a theoretical perspective. We agree that the publication of this work will significantly benefit the community and motivate further theoretical/computational studies.

Reply to Reviewer 4

Reviewer comment 4.0: *This paper studies the interfacial water structure on the HOPG using a combination of 3D-AFM and SEIRAS. While 3D-AFM has greatly advanced our understanding of the liquid-solid interface by providing visualization of the interfacial molecule arrangement, it offers limited information on the physicochemical interactions between them. This paper provides a comprehensive picture of the interfacial water structure on both pristine and hydrocarbon-adsorbed HOPG. The correlative analysis of 3D-AFM and SEIRAS bridges the gap between two contrasting models of the HOPG-water interface: majorly water at the surface and majorly adventitious non-water molecules at the surface. They demonstrate a time- and potential-dependent transition between two states. In addition, they report disrupted hydrogen bonding (HB) at the pristine HOPG-water interface, whereas the hydrocarbon-covered surface shows less broken HB. Given that the crucial roles of interfacial water in physics, biology, and especially chemistry as solvent, reactant, and proton donor to modulate the activation energy of chemical reactions, this well-written paper will be of broad interest to the readership of Nature Communication. The high-quality correlative approach could serve as a valuable framework for future studies of interfacial water structures. We therefore support publication of this manuscript after the authors address the following minor comments.*

Response 4.0: We thank the reviewer for the careful evaluation of this manuscript and for the constructive suggestions. We have addressed the remaining minor comments as detailed below.

Reviewer comment 4.1: *The authors conclude that pristine HOPG, rather than adsorbed hydrophobic molecules (probably adventitious hydrocarbons), weakened the HB network of water molecules leading lower HB-bonded or non-HB-bonded water molecules. Since HOPG itself is a carbon-based materials, what kinds of characteristic of HOPG may contribute to this difference?*

Response 4.1: Indeed, this is an intriguing observation emerging from our data. Both pristine HOPG and adventitious hydrocarbons weaken the hydrogen-bond (HB) network of interfacial water, as reflected by vibrational peak shifts in the SHINERS spectra relative to bulk water. Additionally, our data suggest that pristine HOPG induces a more pronounced HB disruption, resulting in a larger fraction of weakly H-bonded water. While an understanding of the exact origin of this effect will require atomistic simulations that are beyond the scope of this work, here we propose qualitative explanations based on our observations and prior literature.

1. **Structural rigidity/dynamics:** The rigidity and planarity of the HOPG surface may limit the configurational flexibility of nearby water molecules, thereby exacerbating the

disruption of tetrahedral water structures. In contrast, the dynamically moving, less dense interfacial hydrocarbon molecules may be less effective in interrupting the HB network of the surrounding water. This point was already discussed in the original manuscript (originally Page 7 | Lines 9–13, now Page 7 | Lines 21–25): “*This reveals that the hydrocarbon layers are less effective in breaking HBs of nearby water compared to HOPG. ...due to the weaker hydrophobic interaction between water molecules and the dynamically moving, less dense hydrocarbon layers (aged interface) compared to the fixed, densely packed HOPG substrate (pristine interface).*”

2. **Chemical interactions:** Although both HOPG and adventitious hydrocarbons are carbon-based, their physicochemical properties differ substantially. HOPG presents an atomically flat, extended sp^2 carbon lattice with delocalized π electrons, offering a non-polar, highly ordered, and relatively inert surface. In contrast, hydrocarbons, while generally considered nonpolar, contain C–H bonds which possess weak dipole moments due to the electronegativity difference between carbon and hydrogen. This subtle polarity can support dipole–dipole or dipole–induced dipole interactions with nearby water molecules, potentially mitigating HB disruption (*Science* 2021, 374, 1366–1370; *J. Phys. Chem. B* 2023, 127, 3798–3805; *J. Phys. Chem. B* 2002, 106, 2047–2053). We have now added this discussion in the revised manuscript.

Changes to the manuscript: At Page 7 | Lines 25–28 of the main manuscript, we added “*In addition, adventitious hydrocarbons contain C–H bonds with small dipole moments, which can enable weak dipolar interactions with the surrounding water⁶², thus mitigating HB-disruption effects.*”

Reviewer comment 4.2: *The authors should discuss potential zero charge (PZC) of the HOPG? The direction and the strength of the interfacial electric field should be discussed with reference to the PZC (rational potential scale).*

Response 4.2: We thank the reviewer for this thoughtful suggestion. To address this point, we performed electrochemical impedance spectroscopy (EIS) measurements on HOPG in 0.1 M K_2SO_4 (see the added Supplementary Fig. 31). EIS is the most common method for estimating the potential of zero charge (PZC), with the capacitance minimum typically taken as the potential of zero charge.

Our measurements show an expected minimum in the capacitance vs potential profile, consistent with prior studies of graphite in aqueous solutions (*J. Electrochem. Soc.* 1971, 118, 711; *J. Phys. Chem.* 1985, 89, 4249–4251; *Langmuir* 2016, 32, 11448–11455). However, the assignment of this minimum to a well-defined PZC is actually not straightforward.

Early studies by Randin and Yeager (*J. Electrochem. Soc.* 1971, 118, 711; *J. Electroanal. Chem.* 1972, 36, 257–276) and Gerischer (*J. Phys. Chem.* 1985, 89, 4249–4251) highlighted that the graphite electrodes exhibit an unusually low capacitance minimum ($\sim 3\text{--}4 \mu\text{F cm}^{-2}$). They attributed this to the limited electronic density of states at the Fermi level. They argued that this minimum corresponds to a “potential of zero space charge (PZSC)” rather than the true thermodynamic PZC.

Furthermore, PZC is expected to vary depending on the exact interfacial state (pristine vs. non-pristine), as the accumulation of adventitious hydrocarbons at the interface (displacing interfacial water) can easily alter the PZC. A definitive determination of the PZC for these distinct surface states would require dedicated EIS measurements in conjunction with 3D-AFM and SHINERS, which lies beyond the present scope.

Due to the above uncertainties, we refrain from assigning the observed capacitance minimum as a universal PZC at HOPG–aqueous solution interfaces.

Changes to the manuscript:

1) We added a paragraph in the main manuscript (Page 18 | Lines 33–37): “Additionally, electrochemical impedance spectroscopy (EIS) measurements of HOPG in 0.1 M K_2SO_4 solution revealed a minimum differential capacitance at -0.6 V vs Pt (-0.38 V vs Ag/AgCl) (Supplementary Fig. 31). While this value may be close to the potential of zero charge (PZC), the exact PZC value may change at different interfacial states, depending on the presence/absence of adventitious hydrocarbons”

2) We added Supplementary Fig. 31 to the Supplementary Information:

Supplementary Fig. 31 | Electrochemical impedance spectroscopy (EIS) measurements and capacitance analysis. Electrode: HOPG. Electrolyte: 0.1 M K_2SO_4 aqueous solution. **a**, EIS spectra at a series of potentials vs Pt, plotted in complex capacitance plane. Measurements were performed using a Biologic potentiostat over the frequency range of 1 Hz to 1

MHz with an AC perturbation amplitude of 25 mV. The electrode surface area was 0.64 cm^2 . Data in the range of 316 Hz to 1 MHz were used for fitting. Scattered dots correspond to the experimental data and lines are series RC circuit fits. **b**, Capacitance as a function of potential, extracted from RC circuit fits in (a). The capacitance vs potential curve reveals a minimum at -0.6 V vs Pt (-0.38 V vs Ag/AgCl).

Reviewer comment 4.3: I recommend against defining ‘0-0.4 V as OCP’. This is an enormously broad potential range and should not collectively be called OCP.

Response 4.3: We appreciate the reviewer’s rigorous examination of the nomenclature. Our intention was not to redefine OCP, but rather to highlight that across this range no differences can be observed in either 3D-AFM or SHINERS results. Moreover, in the absence of reversible redox reactions to pin it, the OCP of the graphite–aqueous solution interface is not a fixed value. It varies with electrolyte composition and the surface condition of HOPG, and thus we cannot assign a single definitive potential to the large amounts of samples we measured. Nevertheless, to avoid

potential confusions, we have changed the phrases used in the corresponding sentence as described below.

Changes to the manuscript: At Page 3 | Line 28 of the main manuscript, we changed “*are all regarded as OCP*” to “*can be regarded as at or near OCP*”.

Reviewer comment 4.4: *The SHINERS particles will form their own double layer which will potentially be influenced by changes in the charge distribution imparted by the application of different potentials to the working electrode. The authors point to other references that discuss the non-interaction of these particles. However, these discussions need to be present in the current manuscript to make the data credible. Thus, relevant discussion needs to be added.*

Response 4.4: We agree with the reviewer that this is a potential concern that is worth clarification. Despite the inertness of the silica shell, the possibility of double-layer effects from the SHINERS particles must be considered. In fact, this point was already addressed in the original manuscript (originally Page 7 | Lines 15–25 and Supplementary Fig. 17; now Page 7 | Lines 30–41 and Supplementary Fig. 21). Specifically, we wrote:

“In SHINERS measurements, the active interfacial liquid regions contributing to the signal are sandwiched between the HOPG surface and SiO₂ (shell of the Au/SiO₂ particles) (Fig. 1d). To rule out the possible role of silica in producing the emergent interfacial v_{OH} peaks (OH-3 to OH-5), we designed and fabricated a control sample, Au film/SiO₂/electrolyte/SiO₂/Au nanoparticles. As shown in Supplementary Fig. 17, the silica layer is ~3 nm thick at the active gap region. This sample serves as a well-defined control system that enables Raman enhancement, is graphite-free, and ensures SiO₂ is the dominant surface composition in contact with the probed volume of aqueous solution. SHINERS spectra of this control sample at multiple different areas and varying electrode potentials did not exhibit OH-3 to OH-5 peaks (Supplementary Fig. 17), distinct from the results obtained for HOPG/solution interfaces. These data prove that the silica shell layer does not directly contribute to the SHINERS peaks of the HOPG–water interface.”

Since our carefully designed control samples described above also contain the electrical double layers formed on the silica shell, we believe the existing control experiments also verified that the silica shell’s double layers are not a concern. As such, we have revised the last sentence in the above-mentioned paragraph to clarify this point.

Changes to the manuscript: Page 7 | Lines 39–41: “*These data prove that neither the silica shell layer nor the electrical double layer formed on the shell contributed measurably to the SHINERS peaks of the HOPG–water interface.*”

Reviewer comment 4.5: *The authors mention that minimal changes are observed with the presence of ions but Figure 2b shows considerable changes in the force curves. These changes need to be more clearly discussed and elaborated.*

Response 4.5: We thank the reviewer for this important observation. Our statement regarding “minimal changes” was specifically referring to the interlayer spacing d_{12} , which remains consistent across different electrolytes. We agree that Figure 2b displays noticeable variations in the force magnitudes and overall curve profiles. However, these variations are not systematically

correlated with ion presence or electrolyte identity. Instead, they primarily arise from variations in external factors such as AFM tip condition, environmental factors and instrument noise (considering the two different locations and four-year time span). This interpretation is supported by our extensive analysis of the large amounts of data sets together (including measurements from both Urbana and Madrid), which shows that the spread in force values is substantially larger than any systematic dependence on electrolyte concentration or ion identity. We have added extensive error analyses and related discussions in the revised manuscript.

Changes to the manuscript:

1) The original statement in the main manuscript Page 4 | Lines 22–23, “*This interpretation is consistent with prior reports that showed the salt-independence of hydration layer spacings in dilute solutions (<1 M)*”, is now changed to (Page 4 | Lines 28–31) “*This interpretation is consistent with prior reports suggesting that hydration layer spacings are largely insensitive to the identity and concentration of the supporting electrolyte in dilute solutions (<1 M), since the number density of water is much higher than that of the ionic species in these solutions*”.

2) We added a new paragraph in the main manuscript to further clarify the error analysis and potential origins of experimental errors (Page 4 | Lines 39–46): “*Additionally, care must be taken to avoid overinterpreting differences in the observed 3D-AFM force curves. Random fluctuations, due to variations in AFM tip condition, environmental factors and instrument noise (considering the two different locations and four-year time span), can result in various error ranges in the experimental observables. For example, average d_{12} values within 2.8–3.3 Å were observed for pristine HOPG–aqueous solution interfaces with different ionic species and concentrations (Supplementary Fig. 15). Force oscillation amplitudes, on the other hand, exhibits larger fluctuations between 0.01–0.1 nN (Supplementary Fig. 16). These variations should not be directly attributed to differences in the electrolytes or ageing/cleanliness conditions.*”

3) We added Supplementary Fig. 15:

Supplementary Fig. 15 | Summary of interlayer distances (d_{12}) obtained from multiple 3D-AFM measurements at the pristine HOPG–aqueous solution interface across different electrolyte compositions (ion types and concentrations) and electrode potentials. For pure water, the 0-min data from all measured datasets (M1, M2, and M3) were used (corresponding to Supplementary Fig. 3a–c). For 0.1 M KCl, M4 (0 min, post-scan aging at 0, 30, 60, 90, 120 min), M5, and M10 (–1.78 V) were used (Supplementary Figs. 3d, 14a, 11a, and 13c). For 0.1 M K_2SO_4 , datasets U4 and U6 were used (Supplementary Figs. 8 and 10). For 0.2 M K_2SO_4 , datasets M8 (–1.28 V, –1.78 V, and post-scan aging at 20, 40, 60 min) and M9 (–1.28 V, –1.78 V) were used (Supplementary Figs. 13a,b and 14b). For 0.01 M K_2SO_4 , datasets U1, U2 (including potential scan 1: –1.6 V, –2.2 V, –2.6 V), U3 (including potential scan 1: –1.8 V), U5, M6, and M7 were used (Supplementary Figs. 5–7, 9, 11b,c, and 12a,b).

4) We added Supplementary Fig. 16:

Supplementary Fig. 16 | Force oscillation amplitude analysis of 3D-AFM datasets. *a*, A representative 3D-AFM dataset (corresponding to Supplementary Fig. 10a, -0.8 V) together with schematics depicting the extraction of the force oscillation amplitude. *b,c*, Summary of measured force oscillation amplitudes from representative 3D-AFM datasets as a function of electrode potential for (b) pristine HOPG–electrolyte interfaces (corresponding to Supplementary Figs. 5–10, 11a, and 11c) and (c) non-pristine interfaces (corresponding to Supplementary Fig. 13) with different electrolyte compositions and concentrations. The overall spread in force values exceeds any systematic dependence on electrolyte concentration or ion identity.

Reviewer comment 4.6: *How do the authors distinguish hydrocarbon adsorption from potential slow formation of oxygen groups on the HOPG surface that could also lead to structural changes? It would have been nice to show measurements where organics are purposefully added to the electrolyte to corroborate the claim that the changes in force curves occur because of organics adsorption. I realize, however, that this may be beyond the scope of the study.*

Response 4.6: We thank the reviewer for this thoughtful comment. The distinction between hydrocarbon adsorption and the slow formation of oxygen-containing functional groups on the HOPG surface is indeed important, as both could in principle lead to changes in interfacial structure.

Importantly, the entire potential range we explored in this work lies within a regime where HOPG is known to remain stable (see Supplementary Fig. 22 as proof). In this range, oxygen groups are not expected to form (*J. Power Sources* 2008, 185, 740–746; *Anal. Chem.* 1993, 65, 1378–1389; *Sci. Rep.* 2016, 6, 22056).

Further, our conclusion on interfacial hydrocarbon accumulation is supported by multiple existing pieces of evidence:

1. **Spectroscopic evidence:** Parallel SHINERS measurements, as discussed in the manuscript, revealed the emergence of C–H stretching modes and other hydrocarbon-related spectral features, consistent with organics rather than covalently bound oxygen groups.

2. **Oscillatory force profile:** At the non-pristine interface, we observed periodic damped oscillations in the force curves, characteristic of layered liquid structures. In contrast, oxygen groups covalently bound to the HOPG surface would be expected to form a single adsorbed solid layer, which cannot produce force oscillations with multiple peaks.
3. **Control experiments with organic solvents:** Previous studies on pure solvent (hexane) at HOPG surface (*Nanoscale* 2021, 13, 5275–5283) indeed showed force oscillations with an interlayer spacing of $\sim 4\text{--}5$ Å, closely matching that of the interfacial State 2 reported in this work.

Taken together, these results strongly support our conclusion that the observed interfacial evolution originates from hydrocarbon accumulation rather than slow oxidative modification of the HOPG surface.

Changes to the manuscript: All of the above points have been discussed in multiple places throughout the manuscript. To further clarify that HOPG is stable in the measured potential range, we added one sentence in the caption of Supplementary Fig. 22: “*No other redox reactions are observed.*”

Reply to Reviewer 5

Reviewer comment 5.0: *I co-reviewed this manuscript with one of the reviewers who provided the listed reports. This is part of the Nature Communications initiative to facilitate training in peer review and to provide appropriate recognition for Early Career Researchers who co-review manuscripts.*

Response 5.0: We would like to thank the reviewer for helping with this process.